# Interactions between atomically dispersed copper and phosphorous species are key for the hydrochlorination of acetylene

Ting Wang[1], Zhao Jiang[1], Qi Tang[1], Bolin Wang[2], Saisai Wang[1], Mingde Yu[1], Renqin Chang[3], Yuxue Yue[1], Jia Zhao [iD] [1✉] & Xiaonian Li[1✉]

Vinyl chloride, the monomer of polyvinyl chloride (PVC), is industrially synthesized via acetylene hydrochlorination. Thereby, easy to sublimate but toxic mercury chloride catalysts are widely used. It is imperative to find environmentally friendly non-mercury catalysts to promote the green production of PVC. Low-cost copper-based catalysts are promising candidates. In this study, phosphorus-doped Cu-based catalysts are prepared. It is shown that the type of phosphorus configuration and the distribution on the surface of the carrier can be adjusted by changing the calcination temperature. Among the different phosphorus species, the formed P-C bond plays a key role. The coordination structure formed by the interaction between P-C bonds and atomically dispersed $Cu^{2+}$ species results in effective and stable active sites. Insights on how P-C bonds activate the substrate may provide ideas for the design and optimization of phosphorus-doped catalysts for acetylene hydrochlorination.

[1] Industrial Catalysis Institute of Zhejiang University of Technology, Hangzhou 310014, People's Republic of China. [2] School of Chemical Engineering, Northeast Electric Power University, Jilin 132012, China. [3] Research Center of Analysis Measurement, Zhejiang University of Technology, Hangzhou 310014, People's Republic of China. ✉email: jiazhao@zjut.edu.cn; xnli@zjut.edu.cn

The vinyl chloride monomer (VCM) used in the production of polyvinyl chloride (PVC) in China is mainly produced through the hydrochlorination of acetylene. At present, the development of low-cost and environmentally friendly non-precious metal catalysts is still attractive. Cu-based catalysts were widely studied at the beginning because of their high activity in vapor-phase hydrogenation reactions[1]. Similarly, the use of Cu-based catalysts in the hydrochlorination of acetylene has also been found to have good activity. Cu-based catalysts are now being widely investigated, but their conversion is lower than that of precious metal catalysts, and their activity and stability need to be enhanced due to the accumulation of metal active centers and the reduction of metal high valence states[2–8]. At present, its catalytic performance can be improved by means of carrier modification and the addition of other metals.

It is known that the support can be modified by the doping of non-metallic elements such as nitrogen, boron, and phosphorus on carbon materials to improve the catalytic performance of certain reactions[9–12]. In recent years, people are more and more interested in phosphorus-doped carbon materials. Chen[13] et al. found that electrocatalytic oxygen reduction reaction (ORR) activity can be improved because more structural defects are formed after the introduction of heteroatom phosphorus. Liu[14,15] and partners found that the prepared phosphorus-doped carbon nanotubes and graphite layers changed the electronic structure of the carbon material due to phosphorus doping, thus exhibiting high ORR performance and electrocatalytic activity. In order to be applied to acetylene hydrochlorination, there are also related studies have appeared. The support of the gold catalyst prepared by Wang[16] et al. was phosphorus-doped carbon with triphenylphosphine as the phosphorus source, which improves the conversion of acetylene and selectivity to vinyl chloride monomer[17–23]. Various characterization results show that the phosphorus group on the support can improve the dispersion of catalytic active sites, prevent the active gold species $Au^{3+}$ and $Au^+$ from reducing to $Au^0$, and can also delay the coking deposition on the catalyst surface[24–27]. Li[2] et al. prepared a phosphorus-doped copper catalyst supported on spherical activated carbon (SAC) with high activity and good stability. Phosphorus doping promotes the dispersion of copper species, enhances the interaction between metal and support, and inhibits the agglomeration of copper species during the acetylene hydrochlorination process. Wang[28–30] et al. reported that the introduction of phosphorus can inhibit the reduction of $Cu^{2+}$ during the reaction and promote the dispersion of active ingredients on the activated carbon support.

Although there have been many studies on phosphorus doping, it has not been investigated which of the different phosphorus species produced during the preparation process can be the most suitable anchoring sites[31–36] to play a key role in acetylene hydrochlorination and how it interacts with copper species. In order to solve this problem, in this study, non-toxic and low-cost 1-hydroxyethylidene-1,1-diphosphonic acid (HEDP, Fig. S1) is used as a phosphorus source to prepare copper catalysts supported on P-doped activated carbon calcined at different temperatures. The proportion and distribution of phosphorus species on the surface of the carrier can be adjusted at different calcination temperatures. Combined with acetylene conversion and characterization analysis, we found that the P-C bond plays a crucial role in the hydrochlorination of acetylene, and the coordination structure formed by the interaction between atomically dispersed $Cu^{2+}$ species and P-C bond is the reason for the better catalytic performance of the catalyst. Meanwhile, density functional theory (DFT) is used to further determine the optimal structure of the active site and reveal the detailed reaction path and evolution of $C_2H_2$ and HCl on the active site.

## Results and discussion

**Catalyst characterization.** Phosphorus is easily doped into the carbon framework during the calcination process after impregnation. The XPS and EDS results listed in Supplementary Table 1, Fig. 1a show that there are a certain amount of copper and phosphorus in this batch of catalysts, indicating that copper is well loaded on the support, and phosphorus is successfully doped in the carbon framework. The copper content in ICP-AES data is similar to XPS and EDS data, which confirms the reliability of the data (Supplementary Table 2).

In addition, nitrogen adsorption and desorption isotherms are used to measure P-doped activated carbons. Supplementary Table 3 lists the specific surface area and pore structure parameters of P-doped activated carbons calcined at different temperatures. The BET surface areas of Cu/PC200, Cu/PC400, Cu/PC600, and Cu/PC800 are 203, 420, 910, 1005 $m^2g^{-1}$, respectively. Compared with the specific surface area of activated carbon of 1204 $m^2g^{-1}$, both the specific surface area and pore volume of activated carbon after phosphorus doping treatment are smaller. The addition of phosphorus elements may fill and block part of the pore of the carrier and occupy some available space. The larger specific surface area and pore volume of the catalysts calcined at 600 and 800 °C may be caused by the thermal decomposition of phosphorus ligand at high temperatures and the reduction of blocked pores. The pore size of phosphorus-doped activated carbon is similar, and it is also relatively close to activated carbon. Previous reports generally agree that a higher specific surface area can expose more active sites to promote the transfer of the substrate, which is conducive to improving the activity. The results of this study are also the same. The carbon carrier calcined at 800 °C has the largest specific surface area, and the corresponding acetylene conversion rate is also the highest among several catalysts. As shown in Fig. 1b, the $N_2$ adsorption-desorption isotherm of PC800 with the best effect is a typical IV-type curve with obvious hysteresis loop characteristics of a mesoporous structure. Supplementary Fig. 2 shows that several other catalysts also have hysteresis loop characteristics.

Figure 1c shows the XRD patterns of each fresh Cu-based catalyst. The amorphous diffraction peaks of the carbon carrier at 25° and 43° correspond to the plane of (002) and (101), respectively[37]. In addition, no other discernible diffraction peaks are detected in the Cu-based catalyst, which means that the copper particle size is below the detection limit of the XRD instrument, or the copper species on the activated carbon support is in an amorphous form[38,39]. It can be speculated that the copper species may be well dispersed on the P-doped carbon support. Supplementary Fig. 3 shows the morphology of each fresh phosphorus-doped catalyst. There are almost no obvious copper nanoparticles on the surface of the catalyst, indicating that the copper species are well dispersed on the support. The result is consistent with the previous XRD spectra analysis.

In addition, further analysis of HAADF-STEM image (Fig. 1d and Supplementary Fig. 4a, c, e) revealed the presence of predominantly highly dispersed isolated Cu species, and it is almost difficult to detect copper nanoparticles. The single-center copper species supported on the carbon support is confirmed to be the active center of acetylene hydrochlorination reaction, indicating that the active component of the catalyst is mostly composed of atomically dispersed copper[40–42]. The element mapping of the catalyst Cu/PC800 (Fig. 1e) reveals that C, P, and Cu elements are uniformly distributed on the surface of the catalyst, verifying the successful doping of phosphorus in the carbon support, as well as the other catalysts (Supplementary Fig. 4b, d, f).

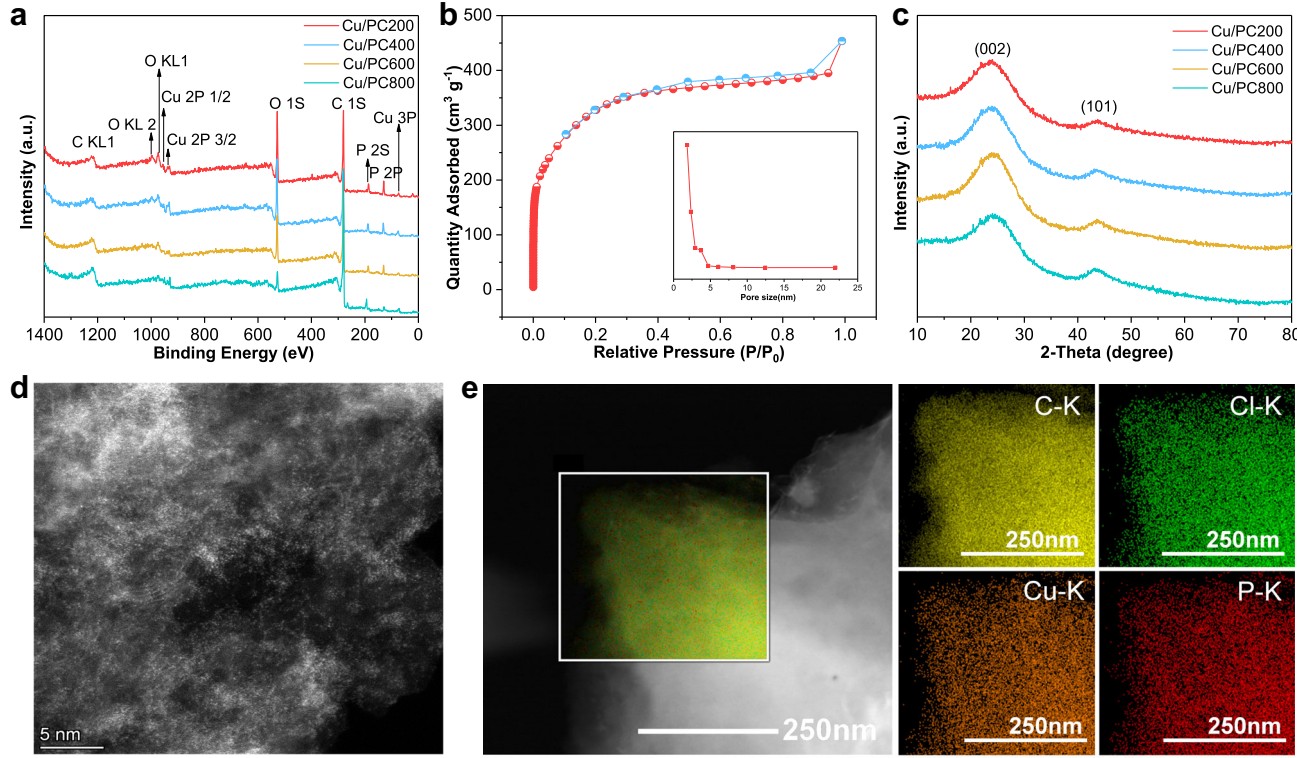

**Fig. 1 Characterizations of catalyst materials. a** Full XPS spectra of fresh P-doped Cu-based catalysts, **b** $N_2$ gas adsorption/desorption isotherms of PC800, **c** XRD pattern of fresh P-doped Cu-based catalysts, **d** Representative HAADF-STEM image of fresh Cu/PC800 catalysts. Representative HAADF-STEM image showing isolated Cu species, **e** EDS elemental mapping of fresh Cu/PC800.

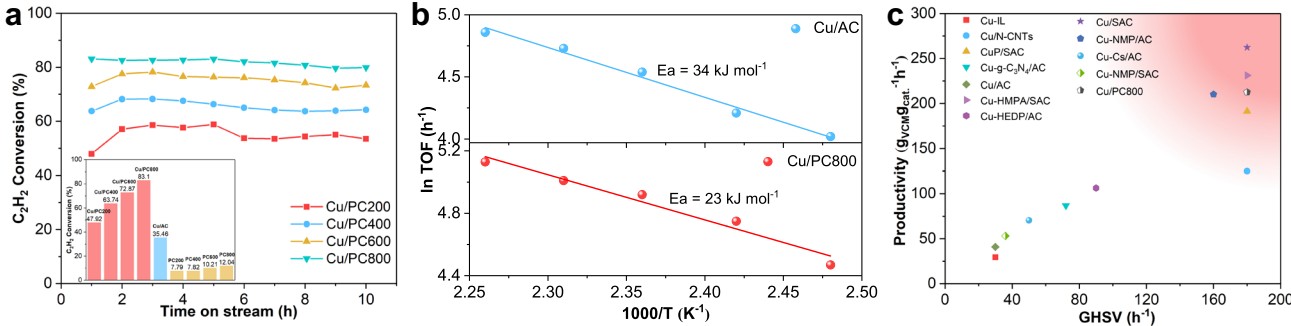

**Fig. 2 Catalysts performance level and kinetic results. a** The conversion of acetylene over P-doped Cu-based catalysts. Reaction conditions: temperature = 150 °C, GHSV($C_2H_2$) = 90 h$^{-1}$, V(HCl)/V($C_2H_2$) = 1.2/1; Comparison of acetylene conversions for Cu/PC200, Cu/PC400, Cu/PC600, and Cu/PC800 catalysts and their respective treated carbons, **b** Kinetic studies of Cu/AC and Cu/PC800 catalyst: apparent activation energy, kJ mol$^{-1}$, **c** GHSV plotted against the Productivity for some copper-based catalysts reported in literature and Cu/PC800 catalyst with better catalytic performance in this article.

**Catalytic performance of Cu-based catalysts**. The catalyst shown in Fig. 2a has the same copper load and phosphorus doping amount in the preparation process. Under the test conditions of T = 150 °C, GHSV($C_2H_2$) = 90 h$^{-1}$ and V(HCl): V($C_2H_2$) = 1.2, the initial conversion of acetylene is significantly different due to the different calcination temperatures of phosphorous doped carbon carriers. The initial conversion of acetylene increases with the increase of calcination temperature. These catalysts don't deactivate within 10 h, and Cu/PC800 shows a better catalytic performance with the highest conversion reaching 83.1%. As we can see, Supplementary Fig. 5 clearly shows that the VCM selectivity of all catalysts has reached more than 99%. Obviously, all the P-doped Cu-based catalysts in Fig. 2a show a higher initial conversion than pure Cu/AC (the initial conversion is 35.46%).

However, the activity of several phosphorus-doped carbon supports without the active component copper is very low, indicating that the enhanced activity of the copper-based catalyst is due to the interaction and synergistic effect between the active copper species and the phosphorus-doped activated carbon, rather than simply the sum of the parts.

In addition, through experiments at different temperatures, the Arrhenius equation is used to plot and the experimental activation energy (Ea) is obtained through linear fitting (Fig. 2b). At this time, the internal and external diffusion of the reaction have been eliminated, and the reaction is under kinetic control. The apparent activation energy of the Cu/PC800 catalyst with the best catalytic performance in the figure is calculated to be 23 kJ mol$^{-1}$. The activation energy of the classic Cu-based

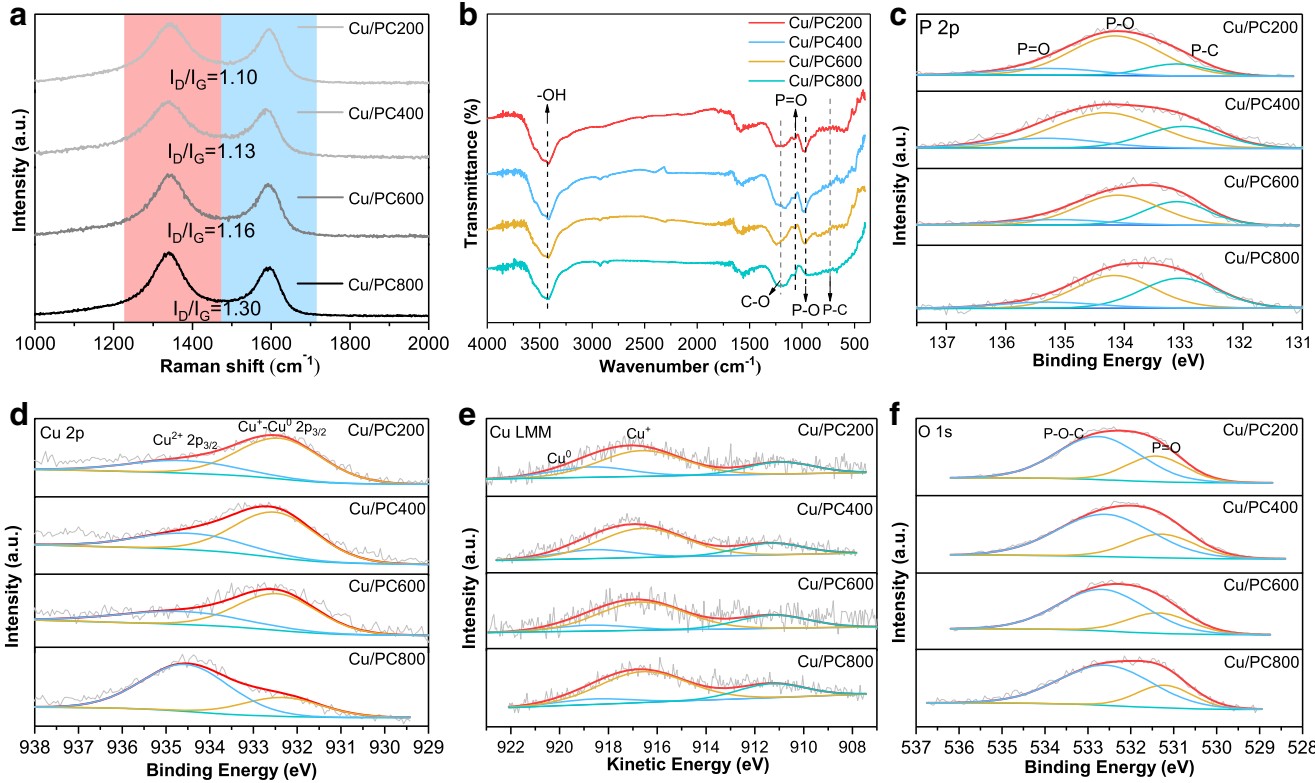

**Fig. 3 Characterization of the P-doped Cu-based catalysts. a** Raman spectra, **b** FT-IR spectra, **c** P 2p XPS spectra, **d** Cu 2p XPS spectra, **e** Cu XAES spectra, and **f** O 1s XPS spectra of fresh P-doped Cu-based catalysts.

catalyst Cu/AC is $34 \, \text{kJ mol}^{-1}$, and the calculated result of the P-doped Cu-based catalyst is lower than this value, indicating that acetylene is more likely to react with hydrogen chloride on the P-doped Cu-based catalyst than the classic copper catalyst. In order to demonstrate the excellent performance of P-doped Cu-based catalysts, the Cu/PC800 catalyst is compared with some copper-based catalysts published in the literature (note that the experimental conditions are not necessarily the same). The productivity at the beginning of the experiment is plotted versus GHSV and is shown in Fig. 2c and Supplementary Table 4. Various copper catalysts from the literature[3,7,43–51] is used for this comparison, and it is obvious that the Cu/PC800 catalyst is one of the better catalysts that can provide higher yields of vinyl chloride under the harsher conditions of relatively high space velocities (colored areas in Fig. 2c).

**Identification of the catalytic active sites**. As shown in Fig. 3a, Raman spectroscopy shows that all samples have two characteristic peaks near 1350 and $1590 \, \text{cm}^{-1}$, which are attributed to the absorption peaks of the D band and G band in the carbon material respectively[7,47]. The G band is generated by the vibration of $sp^2$ hybridized graphite-type carbon atoms, indicating the degree of graphitization of carbon materials, while the D band is usually caused by $sp^3$ hybridized carbon atoms and structural defects, indicating the disordered structure and defect. The doping of phosphorus atoms destroys the hexagonal symmetry of the graphene plane, which increases the number of defect sites in the activated carbon framework. The higher the temperature, the lower the regularity and order of the sample.

XPS spectroscopy can be used to analyze the chemical state of copper on the catalyst surface, and the relative content of different copper species can be calculated according to the relative deconvolution peak area. Figure 3d shows that in each P-doped Cu-based catalyst, the main peak with a binding energy of about

934.5 eV belongs to $Cu^{2+}$, and the peak with a binding energy of about 932.3 eV belongs to $Cu^+$ and $Cu^0$[39,49]. It's worth noting that $Cu^{2+}$ here refers to $CuCl_2$ (around 934.6 eV), rather than CuO whose binding energy position is around 933.6 eV[52–55]. The XAES spectrum can be used to distinguish $Cu^+$ from $Cu^0$. In the XAES spectrum shown in Fig. 3e, the peak of $Cu^+$ can be observed at 916.6 eV, and the peak of $Cu^0$ can be observed at about 918.6 eV[7,47]. The binding energy positions and the relative content of different copper species $Cu^{2+}$, $Cu^+$, and $Cu^0$ are listed (Supplementary Tables 5, 6 and Supplementary Fig. 6). It can be found that metal copper ions are the active component of p-doped Cu-based catalysts, but as the calcination temperature increases, especially when it reaches 800 °C, the ratio of $Cu^{2+}$ increases significantly, and the amount of $Cu^+$ and $Cu^0$ decreases, combined with the result of acetylene conversion (Fig. 4a–c), the activity of the catalyst calcined at 800 °C is the best, so the presence of $Cu^{2+}$ is more conducive to the improvement of catalyst activity.

The P 2p spectrum shown in Fig. 3c can be deconvolved into three peaks to determine the relative content and species of phosphorus. The peaks with binding energies around 135.1, 134.2, and 133.0 eV correspond to three different phosphorus species, P=O, P-O, and P-C respectively[47,50,56], which is also confirmed by FT-IR (Fig. 3b). The strong broadband of the four samples in the range of $3500–3200 \, \text{cm}^{-1}$ corresponds to the -OH stretching vibration, and the broadband around $1300 \, \text{cm}^{-1}$ can be attributed to the C-O stretching vibration[47]. The spectrum clearly shows that the peaks at about $1100 \, \text{cm}^{-1}$ and about $1000 \, \text{cm}^{-1}$ are attributed to the stretching vibration of P=O and P-O, respectively, and some weaker peaks appearing at $750–660 \, \text{cm}^{-1}$ are attributed to P-C[38,50,57]. The P-O bond represents all phosphorus-containing functional groups related to oxygen. The P-C bond indicates that the phosphorus atom is indeed successfully incorporated into the carbon lattice.

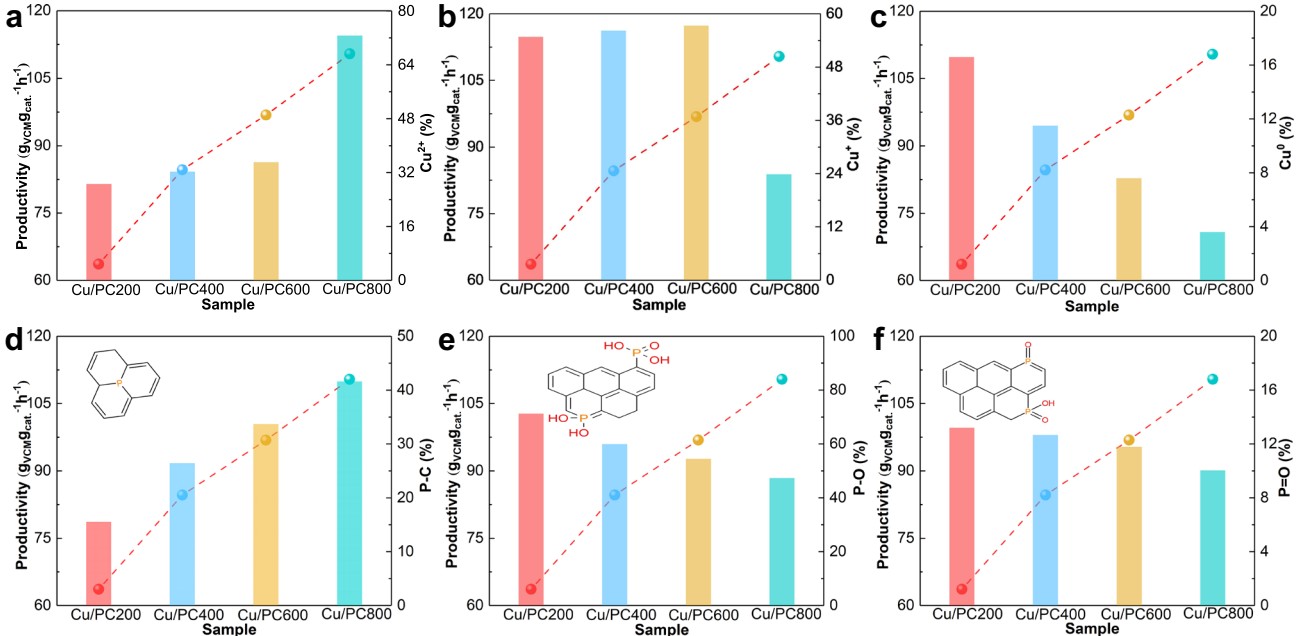

**Fig. 4 Correlation diagram of productivity and various species content.** Correlation diagram of **a**–**c** copper species content, **d**–**f** phosphorus species content and productivity.

The binding energy and the relative amount of each phosphorus species in the p-doped Cu-based catalysts are listed in Supplementary Table 7. Although the P-O bond occupies the highest proportion in each catalyst, only the relative contents of the P-C bond increase significantly with the increase of roasting temperature, while the relative contents of the P=O bond and the P-O bond decrease. Combined with the catalytic activity of Cu-based catalysts (Fig. 4d–f), the P-C bond has the largest proportion (41.6%) in the catalysts calcined at 800 °C, indicating that the higher the content of the P-C bond, the more phosphorus atoms enter the carbon skeleton, the more favorable the catalyst to obtain higher catalytic activity.

In order to further analyze the chemical bond configuration of phosphorus in the catalyst, the O 1 s peak is deconvolved into two components. As shown in Fig. 3f, the peak at about 531.4 eV of all catalysts is attributed to non-bridging oxygen (P = O)[50,58], and the peak at about 532.5 eV of binding energy is attributed to P-O-C bond and accounts for a large proportion[7], which is consistent with the result of P 2p spectrum. It further indicates that the P-O-C bond is included in the oxygen-related phosphorus-containing functional group represented by the P-O bond. In addition, combined with the results of the Supplementary Table 8 and the acetylene conversion (Supplementary Fig. 7), as the calcination temperature increases, the relative content of P=O bond decreases slightly, and the relative content of the P-O-C bond increases slightly, but the change range of both is very small, indicating that the variation of calcination temperature has little effect on the proportion of P=O bond and P-O-C bond and they have no significant positive effect on the activity of the catalyst. It should be noted that although the oxygen-related phosphorus-containing functional groups represented by the P-O bond will decrease with the increase of the calcination temperature, the change of the P-O-C bond contained in it is negligible. In general, the distribution of phosphorus species on the surface of the support can be adjusted by changing the roasting temperature. For the P-doped Cu-based catalyst in this study, the increase of the calcination temperature only leads to a significant increase in the relative content of the P-C bond, which has a positive impact on the catalytic activity. Combined with the results of P 2p and

Cu 2p XPS spectra, it can be inferred that $Cu^{2+}$ species and P-C bond can play a positive role in the hydrochlorination of acetylene, and the coordination structure formed by the interaction between phosphorus species (P-C) and isolated single-atomic $Cu^{2+}$ species is the main active site of the Cu-based catalyst.

It has been reported that experimentally, many phosphorus doping methods, including high-temperature firing in an inert environment, are often accompanied by oxygen doping, so that different types of phosphorus and oxygen-containing functional groups can be formed[59,60]. Referring to the XPS results, P-O bonds, P=O bonds, and P-C bonds may constitute various phosphorus and oxygen-containing groups and the bonding configuration of phosphorus entering the carbon matrix (Fig. 5a).

The TPR curves of each catalyst are shown in Fig. 5b. Two main reduction peaks can be detected in all catalysts. The first hydrogen consumption peak appears in the range of 230 to 330 °C, and the second peak appears in the temperature range of 470 to 570 °C, these two reduction peaks are attributed to the reduction of $Cu^{2+}$ species to $Cu^{+}$ species and the change of $Cu^{+}$ species to metallic copper, respectively[61]. The reduction peaks of $Cu^{2+}$ and $Cu^{+}$ move to higher temperatures with the increase of the calcination temperature of the phosphorus-doped carbon support. The temperatures of the two reduction peaks of Cu/PC800 have increased to different degrees compared with other catalysts. It shows that compared with other catalysts calcined at other temperatures, there is a stronger interaction between copper and phosphorus-doped carbon support in Cu/PC800 catalyst, which effectively improves the anti-reduction ability of $Cu^{2+}$ and $Cu^{+}$ species. In addition, the reduction peak area of $Cu^{2+}$ in the Cu/PC800 catalyst is significantly larger than other catalysts, indicating that the coordination structure formed by Cu and P atoms stabilizes the high-valent copper, and to a certain extent delays the reduction of the oxidation state $Cu^{2+}$ species during the preparation process. This is the same as the result of XPS spectra. The relatively excellent catalytic performance of the Cu/PC800 catalyst comes from the interaction between the $Cu^{2+}$ species with high valence and the P-C bond.

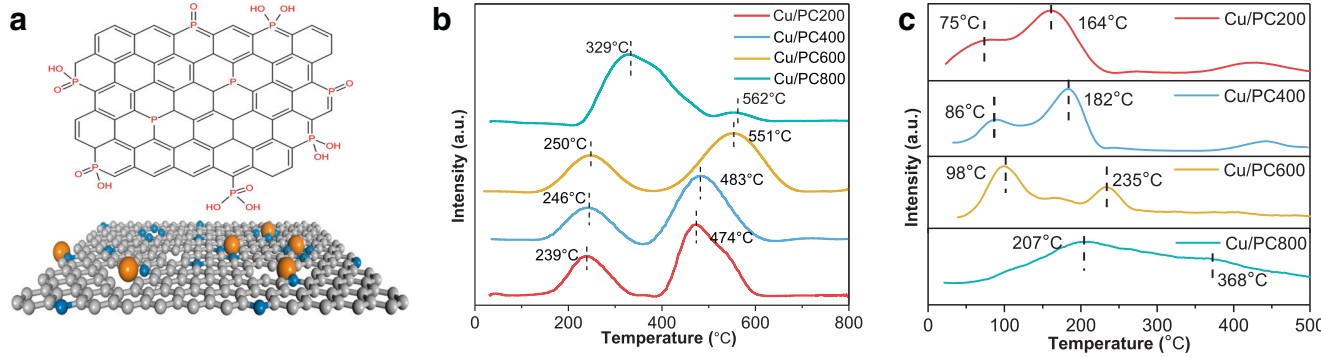

**Fig. 5 Possible phosphorus-doped carbon substrates and adsorption and reduction properties of catalysts. a** Possible structure for P species in fresh P-doped Cu-based catalysts. The gray (carrier), blue and orange balls representing P and Cu atoms, **b** TPR profiles of fresh P-doped Cu-based catalysts, **c** TPD-MS of $C_2H_2$ on Cu/PC200, Cu/PC400, Cu/PC600, and Cu/PC800 catalysts.

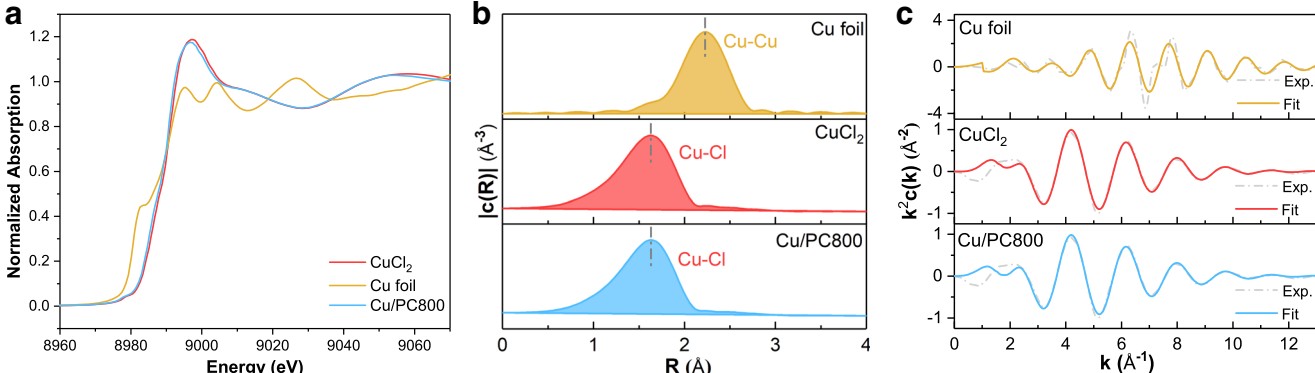

**Fig. 6 Oxidation state and coordination environment of active component Cu. a** Cu K-edge-normalized XANES spectra of the sample and reference material, **b** Fourier-transformed magnitude of Cu foil, $CuCl_2$, and Cu/PC800 (no phase correction). Experimental and fitted EXAFS spectra at the Cu K-edge of the selected catalysts, **c** k-space.

TPD characterization can analyze the adsorption of the catalyst to the reactants, but the phosphorus ligand will be thermally decomposed at a higher temperature. In order to avoid interference to the acetylene signal from the sample desorption, the TPD-MS experiment is used to study the adsorption of acetylene by the active site of the catalyst. Before the TPD-MS analysis, the samples were dried overnight in an oven at 120 °C, and the samples were purged for nearly half an hour before the analysis. The results are shown in Fig. 5c. Each catalyst has two desorption peaks, indicating that there are two active sites capable of adsorbing substrates, which may be $Cu^{2+}$ active centers and $Cu^+$ active centers, respectively. It has been reported that in the Cu(II)/AC catalyst, the adsorption energy of acetylene on the copper center is less than that of the Cu(I)/AC catalyst, and the smaller adsorption energy is usually desorbed first[49]. Therefore, the desorption peak at lower temperature and the desorption peak at higher temperature are likely to be attributed to the active center of $Cu^{2+}$ and $Cu^+$ respectively. The $Cu^{2+}$ active species is the main active center. By comparing the different catalysts, it can be found that with the increase of calcination temperature, the peak area and desorption temperature of the desorption peak related to $Cu^{2+}$ increase, especially for the catalyst with the calcination temperature of 800 °C. The peak area of desorption peak is widely considered to represent the adsorption capacity. The larger the area is, the more active sites exist and the more acetylene is adsorbed on the corresponding active sites. The desorption temperature represents the strength of adsorption, and a high temperature indicates a stronger adsorption capacity for acetylene. Therefore, more $Cu^{2+}$ active sites are conducive to the improvement of the

catalytic performance of our series of phosphorus-doped copper-based catalysts. Some results of TPD-MS are the same as those of XPS and TPR above. Compared with weak adsorption, the relatively strong adsorption of acetylene by the relevant active sites is more beneficial to improve the activity and stability of the catalyst. Nitrogen adsorption and desorption isotherms are used to measure the used Cu/PC800 catalyst. According to the nitrogen adsorption and desorption isotherms, the specific surface area of the catalyst is 986 $m^2\,g^{-1}$, and the pore volume and pore size are 0.57 $cm^{-3}\,g^{-1}$ and 2.14 nm, respectively. The result shows the specific surface area and pore structure parameters of the used catalyst are similar to those of the fresh catalyst during the reaction period of 10 h, and the relatively strong acetylene adsorption does not lead to significant carbon deposition.

Supplementary Fig. 8a shows the XRD pattern of the used sample, which is similar to the fresh catalyst. The two main diffraction peaks at ~25° and 43° correspond to the (002) and (101) crystal planes of carbon, respectively. Except for the two diffraction peaks, no other characteristic peaks were found. The used Cu/PC800 catalyst with the highest catalytic activity was further characterized. The HAADF-STEM image (Supplementary Fig. 8b) shows the presence of highly dispersed isolated copper species. The Cu 2p XPS spectrum (Supplementary Fig. 8c) shows that the peak representing $Cu^{2+}$ at 934.7 eV dominates the spectrum, accounting for 70%. As mentioned above, according to the nitrogen adsorption and desorption isotherms, the used Cu/PC800 does not have obvious carbon deposition. It can be seen that Cu/PC800 is relatively stable during the reaction, active copper species is not easy to agglomerate and not easy to reduce,

and the catalyst also has a certain ability to resist carbon deposition.

X-ray absorption spectroscopy (XAS) further confirms the oxidation state and precise coordination structure of the Cu element in Cu/PC800 (Fig. 6a and Supplementary Table 9). The white line height of each sample is shown in Fig. 6a. The white line intensity value of the cationic Cu standard of $Cu^{2+}$ is 1.18, which is close to the measured values in the literature[61]. The Cu/PC800 catalyst shows similar spectral features to reference $CuCl_2$, and its normalized white line intensity value is 1.17, which is close to the white line intensity of the cationic $Cu^{2+}$ standard sample, indicating that the isolated Cu atoms bear a positive charge of +2. This is consistent with the above XPS results. Although XANES analysis provides relevant information concerning Cu speciation, the complexity of the spectra requires additional extended XAFS (EXAFS) analysis to clarify interpretation. Figure 6b shows EXAFS Fourier transforms (FTs) of Cu foil reference, $CuCl_2$ reference, and Cu/PC800. Fourier-transformed R-space curves of the Cu K-edge EXAFS spectra suggest that Cu is predominantly coordinated with Cl atom in Cu/PC800 centered at about 2.17 Å, and the average coordination number is 3.8. What is important is that no Cu-Cu characteristic bonds are detected, indicating the atomic distribution of Cu elements in the catalyst, which confirms the atomic dispersion of Cu shown by HAADF-STEM. Figure 6c indicates that the experimental data of the EXAFS spectrum are well fitted, as shown by the parameter R factor in Supplementary Table 9.

**Mechanistic studies**. In order to understand the activity of $Cu^{2+}$ active sites supported on the phosphorus-doped carbon carrier, we create carbon substrate based on the selected phosphorus source[38], taking into account the five phosphorus species ($C_3P$, P=O, $(OH)_2P=O$, $P(OH)_2$, $(OH)P=O$) mentioned in Fig. 5a, and then carry out a series of theoretical studies aimed at exploring the potential reaction mechanism. Since $(OH)_2P=O$ is unstable, the higher calcination temperature in the preparation process makes it easy to decompose[62], and we will not consider this site in the following calculation process. We obtain the key role of the P-C bond by XPS spectra, indicating that the increase of phosphorus atoms in carbon lattice is more conducive to the improvement of catalyst activity. It should be noted that each of the phosphorus species in the carbon substrate mentioned above, in which each phosphorus atom enters the carbon skeleton, therefore the active site of several different phosphorus species contains the P-C bond. The overall calculation results are shown in Fig. 7 and Supplementary Fig. 9.

As shown in Supplementary Fig. 9, the adsorption energies of $CuCl_2$ with similar configurations on $C_3P$ and P=O substrates are −240.5 and −86.38 kJ/mol, respectively. Cl atoms are located above P in $C_3P$ and P=O, at distances of 2.59 and 2.31 Å, respectively. For the $P(OH)_2$ adsorption geometry, the adsorption energy of $CuCl_2$ on it is −182.21 kJ/mol, and the Cl ligand in the Cu center seems to interact with the H atom of the hydroxyl group, and the corresponding H-Cl bond length of 2.02 Å. The adsorption energy of $CuCl_2$ on $(OH)P=O$ is −70.89 kJ/mol, and the Cu atom tends to bond with the O atom in the P=O functional group. The active copper species have the highest adsorption energy on $C_3P$, indicating a stronger interaction between the Cu center and the substrate. It can be realized that the interaction between the P atom and its nearby Cl atom will affect the potential reaction mechanism of acetylene hydrochlorination. Therefore, the reaction mechanism on the coordination structure formed by $CuCl_2$ and $C_3P$ is studied.

The mechanism details of acetylene hydrochlorination catalyzed by active copper species on $C_3P$ substrates are presented in Fig. 7 through the DFT modeling. To make the figure clearer, we used simple lines instead of CPK modes to simulate graphene

rings by VMD software[63]. The calculated energy profile is shown in Fig. 7a, and corresponding optimized configurations involved are shown in Fig. 7b. In Fig. 7a, the reaction begins with the coordination of $C_2H_2$ with Cu, a metal atom of $CuCl_2$ in active catalyst **a**, forming intermediate **b** with an adsorption energy of −22.32 kJ/mol and a bond length of C≡C of 1.23 Å (the normal bond length of 1.21 Å). Meanwhile, the P-Cl bond length is 2.44 Å, indicating that phosphorus atom in $C_3P$ will have strong electrostatic interaction with Cl atom in Cu-Cl bond, resulting in electron transfer, as evidenced by the Mayer bond index, which will affect the electronic states around Cu atoms and thus affect the adsorption of substrates. The Cu 2p XPS and fitting parameters from the EXAFS spectra of Cu/PC800 all indicate the coordination between Cu and Cl atom, which also verifies the indirect effect of P atoms on Cu. Then the formation of $C_2H_2$ and HCl co-adsorption configuration on catalyst support in **c** shows the H-Cl bond length in HCl is stretched to 1.342 Å slightly longer than the normal bond length of 1.289 Å in free HCl. With acetylene following adsorbed at the Cl atom of $CuCl_2$ due to the electrostatic attraction between H atom of $C_2H_2$ and Cl atom of $CuCl_2$ to form a weakly less stable intermediate **d**, the distance between the Cl atom of HCl and Cu atom of $CuCl_2$ is 3.14 Å and the Mayer bond index is 0.103. Hirshfeld charges of Cu atom and P atom changes from 0.267 to 0.181 and 0.364 to 0.355 apparently, respectively, which proved our speculation to some extent. Then significantly, the H atom of HCl attacks a C atom of acetylene implied with a six-membered ring structure, which consists of HCl, $C_2H_2$, and $CuCl_2$ in **e**. Visibly, the H-Cl bond length in HCl tends to be broken (1.64 Å). Meanwhile, the bonding tendency of the H atom of HCl and C atom of acetylene is evidenced by the distance and the Mayer bond index, which is 1.30 Å and 0.438, respectively. The Cu center becomes a $CuCl_3$ coordination structure because of the substitution of Cl in **e**, as evidenced by the change of Hirshefeld charge on the Cu center. This step requires overall activation energy of 59.85 kJ/mol and leads to the product complex **f**. At last, desorption of the chloroethylene molecule from **f** regenerates the catalyst with weak desorption energy.

## Conclusions

We synthesized copper-based catalysts supported on phosphorus-doped carbon carriers calcined at different temperatures. Phosphorus atoms have a larger atomic radius and lower electronegativity. Doping phosphorus atoms into the carbon framework is more likely to form a twisted configuration, which can provide a better electronic and geometric coordination environment for single-center copper species[64]. The type and distribution of phosphorus configurations on the support can be adjusted by different calcination temperatures. Combined with the characterization and activity, it was found that the coordination structure formed by the P-C bond and atomically dispersed $Cu^{2+}$ species was the effective active site leading to better performance of the catalyst[65–68]. Density functional theory (DFT) calculation confirms that the optimal active site structure is derived from the interaction between $C_3P$ and atomically dispersed $Cu^{2+}$ species, revealing the detailed reaction path and evolution of $C_2H_2$ and HCl at the active site, indicating that phosphorus atom in $C_3P$ will have strong electrostatic interaction with Cl atoms in Cu-Cl bond, which will affect the electronic states around Cu atoms and thus affect the adsorption of substrates. This may provide some ideas for the design and optimization of phosphorus doping catalysts in the future.

## Methods

**Preparation of catalysts**. P-doped Cu-based catalysts supported on AC were synthesized by the impregnation method with the solvent of deionized water.

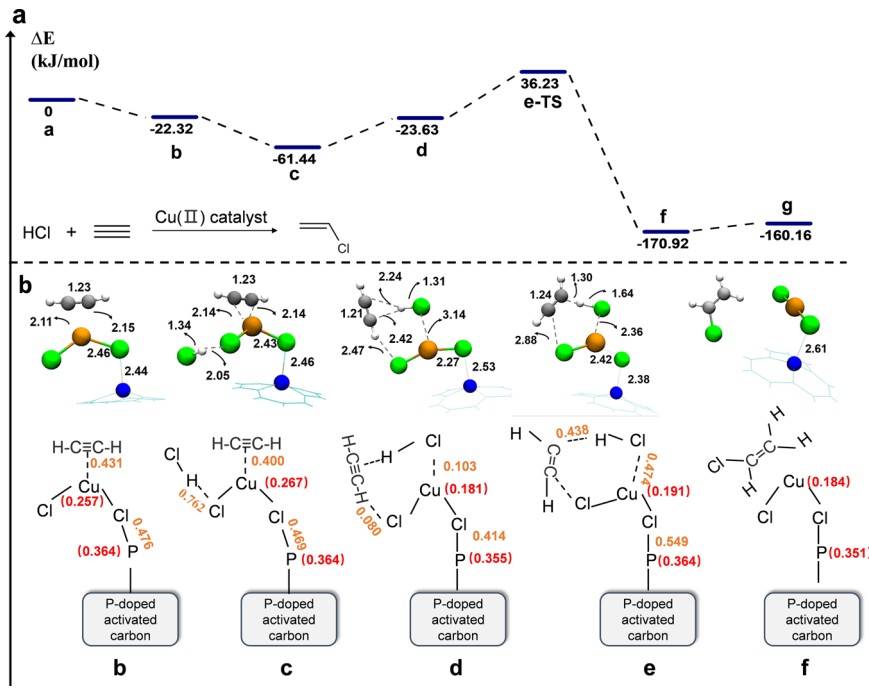

**Fig. 7 DFT calculations on the reaction mechanism. a** Energy profile of acetylene hydrochlorination of HCl and $C_2H_2$, **b** Optimized geometries of intermediates and transition states. Numbers are distances (Å) between atoms, numbers in red are Hirshfeld charges, and numbers in brown are Mayer bond indices.

First, 3.325 g HEDP was dissolved in 10 mL deionized water and stirred intensely until a transparent and clear solution was achieved. About 5 g AC was slowly poured into the above liquid with stirring under room temperature, standing it for 4 h. Then the obtained mixture was dried at 120 °C for 12 h and calcined at 800 °C for 1 h under $N_2$ atmosphere at a heating rate of 10 °C min$^{-1}$ to synthesize carriers named PC800. Supports calcined at 200, 400, and 600 °C were prepared in the same way and named as PC200, PC400, and PC600, respectively. Subsequently, 1.05 g $CuCl_2$ was dissolved in 10 mL deionized water and stirred until a blue homogeneous solution formed. Pretreated carbon PC800 was added to a solution of $CuCl_2$ and maintained at room temperature for 4 h. Finally, the fresh catalyst was acquired after drying at 120 °C for 12 h, denoted as Cu/PC800. By the same method, catalysts calcined at 200, 400, and 600 °C were prepared and named Cu/PC200, Cu/PC400, and Cu/PC600, respectively. In addition, the Cu/AC catalyst for comparison was synthesized by the impregnation method.

**Catalyst characterization.** Transmission electron microscopy (TEM) with a high-angle annular dark-field (HAADF) detector was acquired on an FEI Titan G2 60–300 microscope operating at 300 kV, which can be conducted to detect morphologies of catalysts and to observe the distribution and size of Cu particles. The loadings of Cu were measured using an IRIS (HR) inductively coupled plasma atomic emission spectrometry (ICP-AES). Energy dispersive analysis of X-rays (EDX) was carried out on a Tecnai G2 F30 S transmission electron microscope operating at 300 keV. Nitrogen adsorption/desorption isotherms (BET) were acquired from Micromeritics ASAP 2020 to test various structural parameters of the catalyst. X-ray diffraction (XRD) carried out on a PANalytical-X'Pert PRO generator with Cu Kα radiation was performed to determine dispersity and crystallinity of the active component and crystal form of the support. X-ray photoelectron spectroscopy (XPS) was conducted with a Kratos AXIS Ultra DLD spectrometer to distinguish chemical elements' valence states of the catalyst surface. Raman spectroscopy was carried out in a WITec CRM 200 confocal Raman microscope. Fourier transform infrared spectroscopy (FT-IR) characterization was carried out on a Fourier transform infrared spectrophotometer (Nicolet 6700, Thermal Fisher Nicolet Corporation, Waltham, America). Temperature-programmed reduction (TPR) was conducted in a micro-flow reactor and recorded by a thermal conductivity detector (TCD) to investigate the reduced ability of the catalysts. Temperature-programmed desorption (TPD) was performed in a tubular quartz reactor and recorded by a thermal conductivity detector (TCD). Temperature-programmed desorption (TPD) of the sample and mass spectrometry (MS) analysis of the generated gas were measured on an Omnistar GSD320 mass spectrometer. The extended X-Ray absorption fine structure (EXAFS) spectra was collected at the X-ray absorption spectroscopy (XAS) station of the Shanghai Synchrotron Radiation Facility.

**Computational details.** All structures were optimized using gradient-corrected density functional theory (DFT) with the hybrid B3LYP exchange-correlation

functional, which were performed using Gaussian 09 software package. A 6–31 G(d) basis set was used for all atoms except for Cu, which has been described with Lanl2dz pseudo-potential basis set. All charge and bond-order analyses reported were performed using the NPA charges and Wiberg bond index, respectively. Intrinsic reaction coordinate (IRC) calculations were carried out to confirm the connection of TS to its corresponding reactants and products. $E_{ads}$ followed represent the adsorption energy between the P-doped activated carbon and the catalyst. It was calculated using the following equation:

$$E_{ads} = E_{adsorption-state} - \left(E_{PC} + E_{catalyst}\right)$$

**Catalyst tests.** A fixed-bed glass reactor was used to evaluate the performance of the catalysts. Two mL of the catalyst was added in the fixed position of the tubular reactor. The reactor was flushed with nitrogen for 30 min in order to remove water vapor and air. Then turn on the reaction heating, when the reactor temperature reached 150 °C, $N_2$ was closed and purified hydrogen chloride (HCl) was fed into the tube reactor to activate the catalyst for 30 min. Finally, a reaction gas mixture of acetylene ($C_2H_2$) and hydrogen chloride (HCl) was passed through the reactor at a gas hourly space velocity of $C_2H_2$ (GHSV) of 200 h$^{-1}$ with a $C_2H_2$: HCl ratio of 1:1.2. The effluent streams of the reactor were passed through sodium hydroxide (NaOH) aqueous solution in an absorption bottle to separate any unreacted HCl from the gas products and was sent to be analyzed by an online gas chromatograph equipped with a flame ionization detector (FID) and a Porapak N packed column (6 ft × 1/800 stainless steel).

## Data availability
Any relevant data are available from the authors upon reasonable request.

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

## Acknowledgements

Financial support from the National Natural Science Foundation of China (NSFC; grant No. 21606199), the National Natural Science Foundation of China (NSFC; grant No. 22078302), the Science and Technology Department of Zhejiang Province (LGG20B060004), the China Postdoctoral Science Foundation (2020M671791) and the National Key Research and Development Program of China (2021YFA1501800) are gratefully acknowledged.

## Author contributions

T.W. completed the synthesis, tests, and characterization of catalysts, analyzed and processed most of the characterization data, and completed the article writing. Z.J. carried out the model construction and DFT calculation. Q.T. participated in the catalyst evaluation and completed the format review of the full text. B.W. conceived the idea and conducted partial characterization analysis guidance. S.W. and M.Y. participated in the design and drawing of graphics and tables. R.C. provides Fourier transform infrared spectroscopy (FT-IR) analysis results. Y.Y. helped polish the language of the manuscript. J.Z. and X.L. gave guidance on the full text and supervised the project.

## Competing interests

The authors declare no competing interests.
