## [Peer Review File · Communications Chemistry]

Reviewers' comments:

Reviewer #1 (Remarks to the Author):

A series of phosphorous copper based catalysts were prepared and their catalytic action in acetylene hydrochlorination was analyzed. It provides a reference for the further study of acetylene hydrochlorination. The work load is comprehensive and the catalyst is analyzed by the method of combining experiment and simulation calculation. However, there are some deficiencies in the interpretation of the characterization analysis, and the logic is confused. Therefore, it is suggested to make a major revision of the full text. At the same time, the questions in the attachment need to be answered one by one.

Reviewer #2 (Remarks to the Author):

This manuscript introduces a Cu based single atom catalyst towards hydrochlorination of acetylene. The reaction is environmentally important because mercury was used as the catalyst for this reaction. The results reveals that the doping of phosphorus could form the active site through the interaction between P-C bond and Cu speices, and the role of phosphorus is clearly identified. I support publication after the minor issues listed below are properly addressed.

1. For single-atom catalysts, the characterization procedure is rather fixed. The authors should provide EXAFS of Cu K-edge to confirm the atomic dispersion of Cu.
2. The DFT modeling is way too basic to represent actual catalyst, and the reaction pathway in fig6e should be provided with more convincing results, such as transition state profile.
3. In fig 4, it seem that more than one factor shown is positively correlated with catalytic activity, and the authors should specify which is the most vital one.
4. In addition to XPS, the XANES and soft X-ray absorption spectra should be provide to further illustrate the electronic properties of Cu species.
5. According to Cu LMM Auger spectra, Cu⁰ Cu⁺ Cu²⁺ all exists within the catalyst, and this is contradictory to the single atom structure, because Cu⁰ barely exists in sing atom catalysts.
6. The catalysts after reaction should also be characterized to illustrate the stability issue.
7. Following references regarding atomically dispersed catalysts should be cited. Journal of the American Chemical Society, 2021, 143, 1, 309–317; ACS Catalysis, 2020, 10, 907-913; Nature Nanotechnology, 2019, 14, 354–361.; Journal of the American Chemical Society, 2019, 141, 18921-18925.

Reviewer #3 (Remarks to the Author):

In the process of carrier doping with phosphorus, the author adjusts the types of phosphorus species and their proportions on the surface of the carrier through different calcination temperatures. Combined with the activity, it is determined that the phosphorus species (P-C) plays an important role in the hydrochlorination of acetylene. The interaction between this phosphorus species and atomically dispersed Cu species is beneficial to catalytic activity. Further calculations of density functional theory (DFT) show how the reactants are activated to play a catalytic role. This is a carefully done study and the findings are innovative and insightful. This is fine, because it is already known that atomic dispersed metal sites coordinated on P-doped carbon displayed outstanding catalytic performance and reaction generality in the heterogeneous catalytic reactions (Nature Communications (2020) 11:4074), but little literature has been reported on achieving precise and atomic control to prepare Cu single-atom catalysts for hydrochlorination of acetylene, with excellent catalytic performance as well. In this work, the experimental preparation strategy is ingenious and the mechanism is enlightening, which can certainly contribute to the knowledge of single-atom catalysts for acetylene hydrogenation. This manuscript is recommended for publication after minor revision.

1. How the coordination structure formed by the P-C bond formed after the phosphorus element is incorporated into the carbon lattice and the copper species activates the substrates? The description of this part of the article is too simple, you can describe it in detail.
2. The description of FT-IR in Fig. 3c is not clear enough. It is better to clearly indicate the

wavelength position of each functional group and the corresponding literature source.

3. The author can put the structure of HEDP in the supporting information so that the reader can judge the possible structure of the phosphorus species in the carbon carrier.

Comments

The authors prepared a series of phosphorous doped Cu-based catalysts, and analyzed their catalytic action in acetylene hydrochlorination. The workload was relatively comprehensive, and the catalysts were analyzed by combining experiment and simulation calculation. However, there are some deficiencies in the interpretation of representational analysis and the logic is confused. After careful review, we believe that the following questions need to be clearly explained:

Comment 1: Carefully review the statements in the paper, for example in the abstract, that Cu^{2+} is the main active species from XPS, and that the single atomic copper is the active center as observed in the HAADF-STEM image. And before many people have reported that phosphorus doping can promote catalytic activity in acetylene hydrochlorination reaction, also mentioned in the article, but the author in the abstract and conclusion said that this is the first time to study the role of phosphorus.

Comment 2: At the end of the second paragraph on page 6, the material is a multi-stage pore structure simply by the existence of hysteresis loops in the adsorption and desorption curve. In the XRD analysis process of the third paragraph, the characteristic peak of copper species was not observed in all materials. How can we conclude that the presence of phosphorus improves the dispersion of copper? In Fig. S2, how can we get the structure of phosphorus doped carbon to form two-dimensional nanosheets, and how can we also get that phosphorus doped carbon provides new active sites?

Comment 3: Whether the calculation of TOF value in Fig.2 on page 7 is reasonable, TOF is defined as the amount of reactant converted per unit active site in unit time, not all Cu is active site.

Comment 4: Please confirm whether Table S4 in line 303 of Page 10 should be replaced with Table S5. The abscissa in Fig. 5. (b) is incomplete.

Comment 5: In the TPR analysis on page 11, there is a big difference between the Cu/PC600 pattern and other catalysts, which the author did not explain. Moreover, the author proposed that the coordination structure was formed between Cu and P atoms, which needs to provide more evidence to prove.

Comment 6: In the TPD-MS analysis on page 12, whether the temperature of the first desorption peak is lower than 100 °C is the influence of adsorption of water and gas or physical adsorption, and whether excessive adsorption of acetylene will cause a large amount of carbon deposition.

Comment 7: In the simulation calculation on page 13, P=O structure has a strong adsorption on acetylene, why is the reaction mechanism discussed by using C₃P structure subsequently? Line 411 states that (OH)₂P=O will decompose at reaction temperature, so during the preparation process, the heat treatment temperature is higher than the reaction temperature. Does this structure still exist in the prepared material?

Comment 8: How to determine that phosphorus and carbon are coordinated by C₃P?

Comment 9: There are formatting problems in the PDF version of the manuscript, such as the font of Abstract Acknowledgements and References, as well as the units in the article, etc., which need to be carefully rechecked by the author.

Comment 10: Is the calculation of TOF value in Figure 2 reasonable? TOF value is not given in other literature? The actual amount of metal involved in the reaction cannot be determined.

Dear Reviewers,

On behalf of my co-authors, we thank you very much for giving us an opportunity to revise our manuscript, we appreciate you very much for your positive and constructive comments and suggestions on our manuscript entitled "Isolated Single-Atomic Cu Catalyst Supported on P-doped Carbon for Hydrochlorination of Acetylene". (Manuscript ID: COMMSCHEM-21-0186). We have studied the valuable comments from reviewers carefully, and tried our best to revise the manuscript. The point-to-point response to your comments are listed as following:

Comments in **bold** - Response in black - Actions in yellow

Indicated figures and page numbers refer to the revised manuscript with changes highlighted

Reviewers' comments:

Reviewer #1:

The authors prepared a series of phosphorous doped Cu-based catalysts, and analyzed their catalytic action in acetylene hydrochlorination. The workload was relatively comprehensive, and the catalysts were analyzed by combining experiment and simulation calculation. However, there are some deficiencies in the interpretation of representational analysis and the logic is confused. After careful review, we believe that the following questions need to be clearly explained:

1. Carefully review the statements in the paper, for example in the abstract, that Cu^{2+} is the main active species from XPS, and that the single atomic copper is the

active center as observed in the HAADF-STEM image. And before many people have reported that phosphorus doping can promote catalytic activity in acetylene hydrochlorination reaction, also mentioned in the article, but the author in the abstract and conclusion said that this is the first time to study the role of phosphorus.

Response: Thank you very much for your valuable comments and suggestions. We are sorry that we may not have described it clearly enough in the article. As the reviewer said, many researchers have reported that phosphorus doping can promote the catalytic activity of acetylene hydrochlorination, but the work of this article has carried out further investigations on this basis, and pointed out which phosphorus species can play a key role in the acetylene hydrochlorination reaction, how it interacts with the active Cu species and finally how to activate the substrate. I have made changes in the article. Line 25, “This work is the first to identify which phosphorus species plays a role in the hydrochlorination of acetylene, which may provide some ideas for the design and optimization of phosphorus doping catalysts in the future”, this passage was changed to “Many researchers have reported that phosphorus doping can promote the catalytic activity, but this work further points out the key role of P-C bond, and how it interacts with the active copper species and finally how to activate the substrate, which may provide some ideas for the design and optimization of phosphorus doping catalysts”.

2. At the end of the second paragraph on page 6, the material is a multi-stage pore structure simply by the existence of hysteresis loops in the adsorption and desorption curve. In the XRD analysis process of the third paragraph, the characteristic peak of copper species was not observed in all materials. How can we conclude that the presence of phosphorus improves the dispersion of copper? In Fig. S2, how can we get the structure of phosphorus doped carbon to form two-dimensional nanosheets, and how can we also get that phosphorus doped carbon provides new active sites?

Response: Thanks for the reviewer's correction, we have made correction according to the reviewer's comments. Line 172, the statements of "As shown in the Fig. 1b, the N₂ adsorption-desorption isotherm of PC800 with the best effect is a typical IV type curve with obvious hysteresis loop characteristics of mesoporous structure, indicating that the PC800 carrier has a multi-stage pore structure and other catalysts also have hysteresis loop characteristics" were corrected as "As shown in the Fig. 1b, the N₂ adsorption-desorption isotherm of PC800 with the best effect is a typical IV type curve with obvious hysteresis loop characteristics of mesoporous structure". Line 181, the statements of "It can be speculated that the copper species can be well dispersed on the P-doped carbon support, and the presence of phosphorus can enhance the dispersibility of copper active sites" was deleted as "It can be speculated that the copper species may be well dispersed on the P-doped carbon support". Line 182, the statements of "The Fig. S2 shows the morphology of each phosphorus-doped catalyst, showing classic two-dimensional (2D) carbon nanosheet structure, with obvious

wrinkles which may be caused by phosphorus doping. The size of phosphorus atoms is larger than carbon atoms, resulting in local geometric distortion in the carbon skeleton. Meanwhile, there are almost no copper nanoparticles in these fresh catalysts, indicating that the introduction of phosphorus atoms can make the copper species well dispersed on the support. The presence of phosphorus atoms on the carrier may provide new active sites, and the coordination structure formed with copper species may be the main active sites. These results are consistent with the previous XRD spectra analysis” were corrected as follows: Fig. S3 shows the morphology of each fresh phosphorus-doped catalyst. There are almost no obvious copper nanoparticles on the surface of the catalyst, indicating that the copper species are well dispersed on the support. The result is consistent with the previous XRD spectra analysis.

3. Whether the calculation of TOF value in Fig.2 on page 7 is reasonable, TOF is defined as the amount of reactant converted per unit active site in unit time, not all Cu is active site.

Response: Thanks for the comment of the reviewer, which made me reconsider. To make the data more convincing, we present the GHSV/Productivity diagram by changing TOF to Productivity.

GHSV plotted against the Productivity for some copper-based catalysts reported in literature and Cu/PC800 catalyst with better catalytic performance in this article.

At the same time, we have added Table S7 to the supporting information to record some Cu-based catalysts reported in the literature, which is more convenient for readers to refer to.

Table S7 Comparison of Cu-based catalyst in acetylene hydrochlorination.^a

Catalyst ^c	Catalyst composition ^c			Catalytic performance ^c			Reference ^c
	Active ingredient ^c	Cu/wt.% ^c	Carrier ^c	Reaction conditions ^c	Conversion/% ^c	Selectivity/% ^c	
Cu-g-C ₃ N ₄ ^d /AC ^c	Cu-Pyrrolic N ^c	n.a. ^{a,c}	AC ^c	180°C, 72 h ⁻¹ ^c	79 ^c	>99.5 ^c	3 ^c
Cu-P/SAC ^c	Cu-P ^c	15 ^c	SAC ^c	140°C, 30 h ⁻¹ ^c	99 ^c	99.8 ^c	7 ^c
Cu-IL ^c	Cu ^c	0.7 ^c	n.a. ^{a,c}	180°C, 30 h ⁻¹ ^c	65 ^c	>90 ^c	43 ^c
Cu/N-CNTs ^c	Cu ^c	5.84 ^c	N-CNTs ^c	180°C, 180 h ⁻¹ ^c	47 ^c	>98 ^c	44 ^c
Cu-Cs/AC ^c	Cu-Cs ^c	1 ^c	AC ^c	200°C, 50 h ⁻¹ ^c	92 ^c	>99 ^c	45 ^c
Cu/AC ^c	Cu ^c	5.02 ^c	AC ^c	180°C, 30 h ⁻¹ ^c	>90 ^c	n.a. ^{a,c}	46 ^c
Cu-HEDP ^e /AC ^c	Cu-HEDP ^c	5 ^c	AC ^c	180°C, 90 h ⁻¹ ^c	>80 ^c	>99 ^c	47 ^c
Cu/SAC ^c	Cu ^c	15 ^c	SAC ^c	180°C, 180 h ⁻¹ ^c	98.7 ^c	>99.5 ^c	48 ^c
Cu-NMP ^e /AC ^c	Cu ^c	15 ^c	AC ^c	180°C, 160 h ⁻¹ ^c	>89 ^c	n.a. ^{a,c}	49 ^c
Cu-HMPA ^e /SAC ^c	Cu-HMPA ^c	15 ^c	SAC ^c	180°C, 180 h ⁻¹ ^c	87.25 ^c	>99 ^c	50 ^c
Cu-NMP ^e /SAC ^c	Cu ^c	12 ^c	SAC ^c	180°C, 36 h ⁻¹ ^c	>99.9 ^c	>99.9 ^c	51 ^c
Cu/PC800 ^c	Cu-HEDP ^c	10 ^c	AC ^c	150°C, 180 h ⁻¹ ^c	75 ^c	>99 ^c	This work ^c

^a n.a.: not available.

Combining Table S7 and Fig. 2d, it can be seen that Cu/PC800 can obtain high productivity under the harsh reaction condition of $\text{GHSV}(\text{C}_2\text{H}_2) = 180 \text{ h}^{-1}$, and has a good performance among many reported copper-based catalysts. We made a slight modification in the manuscript, line 231, the statements of “The turnover frequency (TOF) at the beginning of the experiment is plotted versus GHSV, and is shown in Fig. 2d. Various copper catalysts from the literature is used for this comparison, and it is obvious that the Cu/PC800 catalyst is one of the better catalysts that can provide higher yields of vinyl chloride at relatively high space velocities (colored areas in the figure) compared to some of the catalysts reported” were corrected as “The productivity at the beginning of the experiment is plotted versus GHSV, and is shown in Fig. 2d. Various copper catalysts from the literature^{3,7,43-51} is used for this comparison, and it is obvious that the Cu/PC800 catalyst is one of the better catalysts that can provide higher yields of vinyl chloride under the harsher conditions of relatively high space velocities (colored areas in Fig. 2d)”.

4. Please confirm whether Table S4 in line 303 of Page 10 should be replaced with Table S5. The abscissa in Fig. 5. (b) is incomplete.

Response: We are very sorry for our negligence and thank the reviewer for pointing out the mistake. Table S4 has been replaced with Table S5.

Table S5 X-ray photoelectron spectroscopy (XPS) data of Cu XAES spectra of fresh P-doped Cu-based catalysts from this work.^a

Sample ^c	Cu ⁰ ^{a,c}			Cu ⁺ ^{a,c}		
	Position/ ^c	FWHM ^{b/c}	Area/ ^c	Position/ ^c	FWHM ^{b/c}	Area/ ^c
	eV ^c	eV ^c	% ^c	eV ^c	eV ^c	% ^c
Cu/PC200 ^c	918.6 ^c	3.78 ^c	16.6 ^c	916.6 ^c	4.13 ^c	54.8 ^c
Cu/PC400 ^c	918.6 ^c	3.61 ^c	11.5 ^c	916.6 ^c	4.21 ^c	56.2 ^c
Cu/PC600 ^c	918.7 ^c	3.63 ^c	7.6 ^c	916.6 ^c	4.22 ^c	57.3 ^c
Cu/PC800 ^c	918.6 ^c	3.61 ^c	3.6 ^c	916.6 ^c	4.28 ^c	23.8 ^c

^a based on reference values; ^b FWHM = full width at half maximum. ^c

The abscissa in Fig. 5. (b) has been modified completely.

TPD-MS of C₂H₂ on Cu/PC200, Cu/PC400, Cu/PC600 and Cu/PC800 catalysts.

5. In the TPR analysis on page 11, there is a big difference between the Cu/PC600 pattern and other catalysts, which the author did not explain. Moreover, the author proposed that the coordination structure was formed between Cu and P atoms, which needs to provide more evidence to prove.

Response: Thanks a lot for the reviewer's careful pointing out. In the TPR analysis on page 11, the labels of Cu/PC600 and Cu/PC800 were indeed reversed due to the author's carelessness. Cu/PC800 is very different from other catalysts. The green line in the Fig. 5. (a) corresponds to Cu/PC800.

TPR profiles of fresh P-doped Cu-based catalysts.

Thanks to the reviewer for this comment. We're sorry that we may not have described it clearly in the manuscript. The results of Cu 2p XPS and HAADF-STEM show that isolated Cu^{2+} species are the main active components, and P 2p XPS spectra determines the key role of the P-C bond. The original manuscript intends to state that there is an interaction between atomically dispersed Cu^{2+} species, mainly CuCl_2 containing Cu-Cl bonds, and the P-C bond, which can be obtained from the XPS spectra: Line 260, It is worth noting that Cu^{2+} here refers to CuCl_2 (around 934.6 eV), rather than CuO whose binding energy position is around 933.6 eV⁵²⁻⁵⁵. Indeed, as the reviewer said, more evidence is needed to confirm the specific coordination structure of the active site. We have supplemented the X-ray absorption spectra (XAS)

of Cu/PC800 to further confirms the oxidation state and precise coordination structure of the Cu element in Cu/PC800. The results show that Cu in Cu/PC800 mainly exists as Cu^{2+} oxidation state and coordinates with Cl atom. Line 396, “X-ray absorption spectroscopy (XAS) further confirms the oxidation state and precise coordination structure of the Cu element in Cu/PC800. The white line height of each sample is shown in the Fig. 6a. The white line intensity values of cationic Cu standard of Cu^{2+} is 1.18, which is close to the measured values in the literature⁶¹. The Cu/PC800 catalyst shows similar spectral features to reference CuCl_2 , and its normalized white line intensity value is 1.17, which is close to the white line intensity of the cationic Cu^{2+} standard sample, indicating that the isolated Cu atoms bear a positive charge of +2. This is consistent with the above XPS results. Although XANES analysis provides relevant information concerning Cu speciation, the complexity of the spectra requires additional extended XAFS (EXAFS) analysis to clarify interpretation. Fig. 6b shows EXAFS Fourier transforms (FTs) of Cu foil reference, CuCl_2 reference and Cu/PC800. Fourier transformed R-space curves of the Cu K-edge EXAFS spectra suggests that Cu is predominantly coordinated with Cl atom in Cu/PC800 centred at about 2.17 Å, and the average coordination number is 3.8. What is important is that no Cu-Cu characteristic bonds are detected, indicating the atomic distribution of Cu elements in the catalyst, which confirms the atomic dispersion of Cu shown by HAADF-STEM. Fig. 6c indicates that the experimental data of the EXAFS spectrum are well fitted, as shown by the parameter R factor in Table S8” was added.

Table S8 Fitting parameters from the EXAFS spectra of the selected catalysts.

Sample	Scattering Path	CN ^a	R(Å) ^b	$\sigma^2 \times 10^{-3} (\text{Å}^2)$ ^c	R factor (%)
Cu foil	Cu-Cu	12	2.55 ± 0.01	8.9 ± 0.5	0.32
CuCl ₂	Cu-Cl	3.7 ± 0.3	2.16 ± 0.01	11.0 ± 1.1	0.82
Cu/PC800	Cu-Cl	3.8 ± 0.3	2.17 ± 0.01	11.1 ± 1.1	0.84

^a Coordination number, ^b Coordination shell distance, ^c Debye-Waller factor.

(a) Cu K-edge-normalized XANES spectra of the sample and reference material. (b) Fourier-transformed magnitude of Cu foil, CuCl₂ and Cu/PC800 (no phase correction). Experimental and fitted EXAFS spectra at the Cu K-edge of the selected catalysts, (c) k-space.

In addition, DFT calculation results show that phosphorus atoms entering the carbon skeleton will have strong electrostatic interaction with Cl atoms in the Cu-Cl bond, thus affecting the electronic states of Cu atoms, which will affect the adsorption of substrates. Theoretical calculations about this part are presented in the revised manuscript as follows, line 448, “In Fig. 7a, the reaction begins with the coordination of C₂H₂ with Cu, a metal atom of CuCl₂ in active catalyst **a**, forming intermediate **b** with an adsorption energy of -22.32 kJ/mol and a bond length of C≡C of 1.23 Å (the normal bond

length of 1.21 Å). Meanwhile, the P-Cl bond length is 2.44 Å, indicating that phosphorus atom in C₃P will have strong electrostatic interaction with Cl atom in Cu-Cl bond, resulting in electron transfer, as evidenced by the Mayer bond index, which will affect the electronic states around Cu atoms and thus affect the adsorption of substrates. The Cu 2p XPS and fitting parameters from the EXAFS spectra of Cu/PC800 all indicate the coordination between Cu and Cl atom, which also verifies the indirect effect of P atoms on Cu”. In summary, we believe that Cu²⁺ is coordinated with two Cl atoms, and the interaction between P atom in C₃P structure and Cl atom in Cu-Cl bond in CuCl₂ leads to electron transfer and thus affects the electronic states around Cu, which will affect the adsorption and activation of the substrate.

6. In the TPD-MS analysis on page 12, whether the temperature of the first desorption peak is lower than 100 is the influence of adsorption of water and gas or physical adsorption, and whether excessive adsorption of acetylene will cause a large amount of carbon deposition.

Response: Considering the reviewer’s comments, I think it makes sense, “Before the TPD-MS analysis, the samples were dried overnight in an oven at 120°C, and the samples were purged for nearly half an hour before the analysis” was added in line 351. In addition, we want to express that compared with weak adsorption, stronger acetylene adsorption at the active site is more conducive to the improvement of catalytic performance. As for whether our catalyst will cause a large amount of carbon deposition when the acetylene adsorption is relatively strong, we removed the

Cu/PC800 catalyst evaluated for 10h from the reaction tube. According to the nitrogen adsorption and desorption isotherms, the specific surface area of the catalyst is $986 \text{ m}^2\text{g}^{-1}$, and the pore volume and pore size are $0.57 \text{ cm}^3\text{g}^{-1}$ and 2.14 nm , respectively. The result shows that there is no obvious carbon deposition. At the same time, we added relevant content on line 370 of the manuscript: Compared with weak adsorption, the relatively strong adsorption of acetylene by the relevant active sites is more beneficial to improve the activity and stability of the catalyst. Nitrogen adsorption and desorption isotherms are used to measure the used the Cu/PC800 catalyst. According to the nitrogen adsorption and desorption isotherms, the specific surface area of the catalyst is $986 \text{ m}^2 \text{ g}^{-1}$, and the pore volume and pore size are $0.57 \text{ cm}^3 \text{ g}^{-1}$ and 2.14 nm , respectively. The result shows the specific surface area and pore structure parameters of the used catalyst are similar to those of the fresh catalyst during the reaction period of 10h, and the relatively strong acetylene adsorption does not lead to significant carbon deposition.

7. In the simulation calculation on page 13, P=O structure has a strong adsorption on acetylene, why is the reaction mechanism discussed by using C₃P structure subsequently? Line 411 states that (OH)₂P=O will decompose at reaction temperature, so during the preparation process, the heat treatment temperature is higher than the reaction temperature. Does this structure still exist in the prepared material?

Response: Thank you very much for this question raised by the reviewer. As for the theoretical calculation, the original modeling is too basic and the substrate is too simple, so we refer to the updated literature (Inorganic Chemistry Frontiers (2019) 6: 2944-2952) to recreate a more realistic substrate model during modeling, as shown in the figure below. By DFT calculation, when the phosphorus atom replaces a carbon atom in the carbon skeleton and presents a C₃P configuration, the energy released when it interacts with Cu²⁺ species to form coordination structure is the largest, indicating that the interaction between active copper species and C₃P structure is the strongest. The adsorption energy of Cu²⁺ species on several other structures is relatively small, including the P=O structure. As presented in the manuscript, Line 432, “As shown in Fig. S9, the adsorption energies of CuCl₂ with similar configurations on C₃P and P=O substrates are -240.5 kJ/mol and -86.38 kJ/mol, respectively. Cl atoms are located above P in C₃P and P=O, at distances of 2.59 Å and 2.31 Å, respectively. For the P(OH)₂ adsorption geometry, the adsorption energy of CuCl₂ on it is 182.21 kJ/mol, and the Cl ligand in the Cu center seems to interact with the H atom of the hydroxyl group, and the corresponding H-Cl bond length of 2.02 Å. The adsorption energy of CuCl₂ on (OH)P=O is -70.89 kJ/mol, and the Cu atom tends to bond with the O atom in the P=O functional group. The active copper species have the highest adsorption energy on C₃P, indicating a stronger interaction between the Cu center and the substrate. It can be realized that the interaction between the P atom and its nearby Cl atom will affect the potential reaction mechanism of acetylene hydrochlorination. Therefore, the reaction mechanism on the coordination structure

formed by CuCl_2 and C_3P is studied". Therefore, on this basis, we have obtained that phosphorus and carbon are coordinated by C_3P , and the reaction mechanism is discussed by using C_3P structure subsequently, as shown in the figure below.

Optimized geometries of Cu^{2+} active sites adsorption on (a) C_3P , (b) $\text{P}=\text{O}$, (c) $\text{P}(\text{OH})_2$, (d) $(\text{OH})\text{P}=\text{O}$,

respectively. Numbers in black are distances (\AA) between atoms.

Thanks for the reviewer's comments. We are really sorry for our negligence on this issue. Considering the reviewer's suggestion, we have made corrections. The original line 411, "The $(\text{OH})_2\text{P}=\text{O}$ would decompose at our reaction temperature" was deleted. In addition, line 423, "Since $(\text{OH})_2\text{P}=\text{O}$ is unstable, the higher calcination temperature in the preparation process makes it easy to decompose⁶², and we will not consider this site in the following calculation process" was added.

8. How to determine that phosphorus and carbon are coordinated by C_3P ?

Response: Thanks for the reviewer's comment. We refer to a literature on phosphorus-doped carbon materials (ACS Nano (2016) 10: 2305-2315), which

shows a schematic diagram of the phosphorus-doped carbon substrate, as shown in the figure below.

Schematic diagram of the phosphorus-doped carbon substrate.

We refer to this literature to create a carbon substrate according to the selected phosphorus source. Thank you very much for the question raised by the reviewers, which made me rethink. The article may not be clearly stated. We obtained the key role of P-C bond through XPS spectra, indicating that the increase of phosphorus atoms in the carbon lattice is more conducive to the improvement of catalyst activity. The carbon substrate was created by considering five different sites, in which each phosphorus atom entered the carbon skeleton, so that the active sites of several different phosphorus species contained the P-C bond, as shown below.

Possible structures for P species in fresh P-doped Cu-based catalysts. The gray (carrier), blue and orange balls representing P and Cu atoms.

Our original substrate C₁₃H₉ referred to the literature (RSC Advances (2015) 5: 104071-104078), and the previously selected substrate may be too simple, so we have re-referred to the updated literature (Inorganic Chemistry Frontiers (2019) 6: 2944-2952), and selected a larger and more realistic substrate for modeling. By DFT calculation, when the phosphorus atom replaces a carbon atom in the carbon skeleton and presents a C₃P configuration, the energy released when it interacts with Cu²⁺ species to form coordination structure is the largest, indicating that the interaction between active copper species and C₃P structure is the strongest. Therefore, on this basis, we have obtained that phosphorus and carbon are coordinated by C₃P, as shown in the figure below.

Optimized geometries of Cu^{2+} active sites adsorption on (a) C_3P , (b) $P=O$, (c) $P(OH)_2$, (d) $(OH)P=O$,

respectively. Numbers in black are distances (\AA) between atoms.

We have rewritten this part according to the reviewer's comments. Line 419, the statement of "To understand the activity of Cu^{2+} active sites loaded on phosphorus-doped carbon support, considering five phosphorus species (C_3P , $P=O$, $(OH)_2P=O$, $P(OH)_2$, $(OH)P=O$) mentioned in Fig. 4d-f, we carry out a series of theoretical studies aimed at exploring the underlying reaction mechanism" was corrected as "In order to understand the activity of Cu^{2+} active sites supported on the phosphorus-doped carbon carrier, we create carbon substrate based on the selected phosphorus source³⁸, taking into account the five phosphorus species (C_3P , $P=O$, $(OH)_2P=O$, $P(OH)_2$, $(OH)P=O$) mentioned in Fig. 5c, and then carry out a series of theoretical studies aimed at exploring the potential reaction mechanism. Since $(OH)_2P=O$ is unstable, the higher calcination temperature in the preparation process makes it easy to decompose⁶², and we will not consider this site in the following calculation process. We obtain the key role of P-C bond by XPS spectra, indicating that the increase of phosphorus atoms in

carbon lattice is more conducive to the improvement of catalyst activity. It should be noted that each of the phosphorus species in the carbon substrate mentioned above, in which each phosphorus atom enters the carbon skeleton, therefore the active site of several different phosphorus species contains the P-C bond". Line 432, the statement of "The adsorption energy of C₂H₂ on five different phosphorus species active sites are -31.77, -42.27, -85.07, -21.26, -16.54kJ/mol respectively. It is obvious that P(OH)₂ and (OH)P=O copper active sites show weak interaction with C₂H₂. The P=O active site shows better adsorption to C₂H₂, but O of P=O bond may interact strongly with the Cl of HCl, which may cause complex reaction mechanism. Here we do not discuss the P=O species. The (OH)₂P=O would decompose at our reaction temperature. Then only the C₃P specie was left" was modified to "As shown in Fig. S9, the adsorption energies of CuCl₂ with similar configurations on C₃P and P=O substrates are -240.5 kJ/mol and -86.38 kJ/mol, respectively. Cl atoms are located above P in C₃P and P=O, at distances of 2.59 Å and 2.31 Å, respectively. For the P(OH)₂ adsorption geometry, the adsorption energy of CuCl₂ on it is 182.21 kJ/mol, and the Cl ligand in the Cu center seems to interact with the H atom of the hydroxyl group, and the corresponding H-Cl bond length of 2.02 Å. The adsorption energy of CuCl₂ on (OH)P=O is -70.89 kJ/mol, and the Cu atom tends to bond with the O atom in the P=O functional group. The active copper species have the highest adsorption energy on C₃P, indicating a stronger interaction between the Cu center and the substrate. It can be realized that the interaction between the P atom and its nearby Cl atom will affect the potential reaction mechanism of acetylene hydrochlorination.

Therefore, the reaction mechanism on the coordination structure formed by CuCl_2 and C_3P is studied”.

9. There are formatting problems in the PDF version of the manuscript, such as the font of Abstract, Acknowledgements and References, as well as the units in the article, etc., which need to be carefully rechecked by the author.

Response: Thank you very much for the suggestions and reminders of the reviewer. We have rechecked the format of the article and adjusted it. The number of words in the abstract has been revised to meet the journal's requirements (150 words limit).

As an environmentally friendly catalyst for the hydrochlorination of acetylene, Cu-based catalysts have always attracted attention. In this study, a series of phosphorus-doped Cu-based catalysts were prepared by the impregnation method. The type of phosphorus configuration and the distribution on the surface of the carrier can be adjusted by changing the calcination temperature. Among the different phosphorus species, P-C bond formed after the phosphorus element is incorporated into the carbon lattice plays a key role. The coordination structure formed by the interaction between P-C bond and atomically dispersed Cu^{2+} species is effective and stable active site. Many researchers have reported that phosphorus doping can promote the catalytic activity, but this work further points out the key role of P-C bond, and how it interacts with the active copper species and finally how to activate the substrate, which may provide some ideas for the design and optimization of phosphorus doping catalysts.

Abstract.

The acknowledgement format has been modified, as shown below.

Acknowledgements

Financial support from the National Natural Science Foundation of China (NSFC; grant No. 21606199), the Science and Technology Department of Zhejiang Province (LGG20B060004) and the China Postdoctoral Science Foundation (2020M671791) are gratefully acknowledged.

Acknowledgements.

The format of the reference has been modified, which now meets the requirements of the journal.

1. Kobayashi, H., Takezawa, N. & Minochi, C. Methanol-reforming reaction over copper-containing catalysts — The effects of anions and copper loading in the preparation of the catalysts by kneading method. *J. Catal.* **69**, 487-494 (1981).
2. Ren, Y. et al. Chlorocuprate(I) ionic liquid as an efficient and stable Cu-based catalyst for hydrochlorination of acetylene. *Catal. Sci. Technol.* **9**, 2868-2878 (2019).

The format of the references in the manuscript.

References

Communications Chemistry uses standard *Nature* referencing style.

Published papers:

Printed journals

Schott, D. H., Collins, R. N. & Bretscher, A. Secretory vesicle transport velocity in living cells depends on the myosin V lever arm length. *J. Cell Biol.* **156**, 35-39 (2002).

Online only

Bellin, D. L. et al. Electrochemical camera chip for simultaneous imaging of multiple metabolites in biofilms. *Nat. Commun.* **7**, 10535; 10.1038/ncomms10535 (2016).

For papers with more than five authors include only the first author's name followed by 'et al.'.

Reference format required by the journal.

The unit problem in the original manuscript has been revised.

synthesize carriers named PC800. Supports calcined at 200 °C, 400 °C and 600 °C were prepared in the same way and named as PC200, PC400 and PC600, respectively.

3.2 Catalytic performance of Cu-based catalysts

The catalyst shown in the Fig. 2a has the same copper load and phosphorus doping amount in the preparation process. Under the test conditions of $T = 150\text{ }^{\circ}\text{C}$, is the best when calcination temperature is 800 °C, so the presence of Cu^{2+} is more conducive to the improvement of catalyst activity.

Unit problem in the original manuscript.

and calcined at 800 °C for 1h under N_2 atmosphere at a heating rate of $10\text{ }^{\circ}\text{C min}^{-1}$ to

synthesize carriers named PC800. Supports calcined at 200 °C, 400 °C and 600 °C were

The catalyst shown in the Fig. 2a has the same copper load and phosphorus doping amount in the preparation process. Under the test conditions of $T = 150\text{ }^{\circ}\text{C}$, GHSV(C_2H_2)

result of acetylene conversion (Fig. 4a-c), the activity of the catalyst calcined at 800 °C is the best, so the presence of Cu^{2+} is more conducive to the improvement of catalyst

Units in the revised manuscript.

10. Is the calculation of TOF value in Figure 2 reasonable? TOF value is not given in other literature? The actual amount of metal involved in the reaction cannot be determined.

Response: Special thanks to the reviewer for this opinion, which made me rethink. As the reviewer said, the TOF value was not given in other literatures, and the data in the original manuscript were calculated by ourselves. Considering the question raised by the reviewer, and in order to make the data more convincing, we present the GHSV/Productivity diagram by changing TOF to Productivity.

GHSV plotted against the Productivity for some copper-based catalysts reported in literature and

Cu/PC800 catalyst with better catalytic performance in this article.

In addition, we have added Table S7 to the supporting information to record some Cu-based catalysts reported in the literature, which is more convenient for readers to refer to.

Table S7 Comparison of Cu-based catalyst in acetylene hydrochlorination.[†]

Catalyst [†]	Catalyst composition [†]			Catalytic performance [†]			Reference [†]
	Active ingredient [†]	Cu/wt.% [†]	Carrier [†]	Reaction conditions [†]	Conversion/% [†]	Selectivity /% [†]	
Cu-g-C ₃ N ₄ [†] /AC [†]	Cu-Pyrrolic N [†]	n.a. ^{a,†}	AC [†]	180°C, 72 h ⁻¹ [†]	79 [†]	>99.5 [†]	3 [†]
Cu-P/SAC [†]	Cu-P [†]	15 [†]	SAC [†]	140°C, 30 h ⁻¹ [†]	99 [†]	99.8 [†]	7 [†]
Cu-IL [†]	Cu [†]	0.7 [†]	n.a. ^{a,†}	180°C, 30 h ⁻¹ [†]	65 [†]	>90 [†]	43 [†]
Cu/N-CNTs [†]	Cu [†]	5.84 [†]	N-CNTs [†]	180°C, 180 h ⁻¹ [†]	47 [†]	>98 [†]	44 [†]
Cu-Cs/AC [†]	Cu-Cs [†]	1 [†]	AC [†]	200°C, 50 h ⁻¹ [†]	92 [†]	>99 [†]	45 [†]
Cu/AC [†]	Cu [†]	5.02 [†]	AC [†]	180°C, 30 h ⁻¹ [†]	>90 [†]	n.a. ^{a,†}	46 [†]
Cu-HEDP [†] /AC [†]	Cu-HEDP [†]	5 [†]	AC [†]	180°C, 90 h ⁻¹ [†]	>80 [†]	>99 [†]	47 [†]
Cu/SAC [†]	Cu [†]	15 [†]	SAC [†]	180°C, 180 h ⁻¹ [†]	98.7 [†]	>99.5 [†]	48 [†]
Cu-NMP [†] /AC [†]	Cu [†]	15 [†]	AC [†]	180°C, 160 h ⁻¹ [†]	>89 [†]	n.a. ^{a,†}	49 [†]
Cu-HMPA [†] /SAC [†]	Cu-HMPA [†]	15 [†]	SAC [†]	180°C, 180 h ⁻¹ [†]	87.25 [†]	>99 [†]	50 [†]
Cu-NMP [†] /SAC [†]	Cu [†]	12 [†]	SAC [†]	180°C, 36 h ⁻¹ [†]	>99.9 [†]	>99.9 [†]	51 [†]
Cu/PC800 [†]	Cu-HEDP [†]	10 [†]	AC [†]	150°C, 180 h ⁻¹ [†]	75 [†]	>99 [†]	This work [†]

^a n.a.: not available.[†]

Combining Table S7 and Fig. 2d, it can be seen that Cu/PC800 can obtain high productivity under the harsh reaction condition of $\text{GHSV}(\text{C}_2\text{H}_2) = 180 \text{ h}^{-1}$, and has a good performance among many reported copper-based catalysts. We made a slight modification in the manuscript, line 231, the statements of “The turnover frequency (TOF) at the beginning of the experiment is plotted versus GHSV, and is shown in Fig. 2d. Various copper catalysts from the literature is used for this comparison, and it is obvious that the Cu/PC800 catalyst is one of the better catalysts that can provide higher yields of vinyl chloride at relatively high space velocities (colored areas in the figure) compared to some of the catalysts reported” were corrected as “**The productivity at the beginning of the experiment is plotted versus GHSV, and is shown in Fig. 2d. Various copper catalysts from the literature^{3,7,43-51} is used for this comparison, and it is obvious that the Cu/PC800 catalyst is one of the better catalysts that can provide higher yields of vinyl chloride under the harsher conditions of relatively high space velocities (colored areas in Fig. 2d)**”.

Thank you again for your suggestions. All your suggestions are very important, and they have important guiding significance for my future scientific research work.

Reviewer #2:

This manuscript introduces a Cu based single atom catalyst towards hydrochlorination of acetylene. The reaction is environmentally important because mercury was used as the catalyst for this reaction. The results reveals that the doping of phosphorus could form the active site through the interaction between P-

C bond and Cu species, and the role of phosphorus is clearly identified. I support publication after the minor issues listed below are properly addressed.

1. For single-atom catalysts, the characterization procedure is rather fixed. The authors should provide EXAFS of Cu K-edge to confirm the atomic dispersion of Cu.

Response: The reviewer's suggestion is very useful. In this paper, we have proved that Cu has high dispersion on P-doped carbon carrier by XRD and TEM, and further found that the active component is mainly composed of atomically dispersed Cu by HAADF-STEM image, as shown in the figure below.

XRD pattern of fresh P-doped Cu-based catalysts.

HRTEM image of (a) Cu/PC200, (b) Cu/PC400, (c) Cu/PC600 and (d) Cu/PC800 catalysts.

Representative HAADF-STEM image of fresh (a) Cu/PC200, (b) Cu/PC400, (c) Cu/PC600 and (d) Cu/PC800 catalysts.

As the reviewer said, the characterization process is quite fixed for single-atom catalysts, so we provide EXAFS of Cu K-edge to confirm the atomic dispersion of Cu. We have supplemented X-ray absorption spectra (XAS) of Cu/PC800 with the highest activity to further confirm the oxidation state and precise coordination structure of Cu elements. The results show that Cu in Cu/PC800 mainly exists in the oxidation state of Cu^{2+} and coordinates with Cl atoms, and no Cu-Cu bond is detected, which confirms the atomic dispersion of Cu. The supplementary content in the manuscript is as follows, line 396, “X-ray absorption spectroscopy (XAS) further confirms the oxidation state and precise coordination structure of the Cu element in Cu/PC800. The white line height of each sample is shown in the Fig. 6a. The white line intensity values of cationic Cu standard of Cu^{2+} is 1.18, which is close to the measured values in the literature⁶¹. The Cu/PC800 catalyst shows similar spectral features to reference CuCl_2 , and its normalized white line intensity value is 1.17, which is close to the white line intensity of the cationic Cu^{2+} standard sample, indicating that the isolated Cu atoms bear a positive charge of +2. This is consistent with the above XPS results. Although XANES analysis provides relevant information concerning Cu speciation, the complexity of the spectra requires additional extended XAFS (EXAFS) analysis to clarify interpretation. Fig. 6b shows EXAFS Fourier transforms (FTs) of Cu foil reference, CuCl_2 reference and Cu/PC800. Fourier transformed R-space curves of the Cu K-edge EXAFS spectra suggests that Cu is predominantly coordinated with Cl atom in Cu/PC800 centred at about 2.17 Å, and the average coordination number is 3.8. What is important is that no Cu-Cu characteristic bonds are detected, indicating

the atomic distribution of Cu elements in the catalyst, which confirms the atomic dispersion of Cu shown by HAADF-STEM. Fig. 6c indicates that the experimental data of the EXAFS spectrum are well fitted, as shown by the parameter R factor in Table S8” was added.

Table S8 Fitting parameters from the EXAFS spectra of the selected catalysts.[†]

Sample [†]	Scattering Path [†]	CN ^a [†]	R(Å) ^b [†]	$\sigma^2 \times 10^{-3} (\text{Å}^2)$ ^c [†]	R factor (%) [†]
Cu foil [†]	Cu-Cu [†]	12 [†]	2.55 ± 0.01 [†]	8.9 ± 0.5 [†]	0.32 [†]
CuCl ₂ [†]	Cu-Cl [†]	3.7 ± 0.3 [†]	2.16 ± 0.01 [†]	11.0 ± 1.1 [†]	0.82 [†]
Cu/PC800 [†]	Cu-Cl [†]	3.8 ± 0.3 [†]	2.17 ± 0.01 [†]	11.1 ± 1.1 [†]	0.84 [†]

^a Coordination number. ^b Coordination shell distance. ^c Debye-Waller factor.[†]

(a) Cu K-edge-normalized XANES spectra of the sample and reference material. (b) Fourier-transformed magnitude of Cu foil, CuCl₂ and Cu/PC800 (no phase correction). Experimental and fitted EXAFS spectra at the Cu K-edge of the selected catalysts, (c) k-space.

2. The DFT modeling is way too basic to represent actual catalyst, and the reaction pathway in fig6e should be provided with more convincing results, such as transition state profile.

Response: Thanks to the reviewer for the suggestion. Our original substrate $C_{13}H_9$ refers to the literature (RSC Advances (2015) 5: 104071-104078), and the carbon substrate in the literature is shown below.

However, as the reviewer said, the original substrate may be too simple, so we refer to the updated literature again (Inorganic Chemistry Frontiers (2019) 6: 2944-2952), a larger and more realistic substrate is selected for re-calculation during modeling, as shown in the figure below. By DFT calculation, when the phosphorus atom replaces a carbon atom in the carbon skeleton and presents a C_3P configuration, the energy released when it interacts with Cu^{2+} species to form coordination structure is the largest, indicating that the interaction between active copper species and phosphorus-doped carbon carrier is the strongest. Therefore, we have obtained that phosphorus and carbon are coordinated by C_3P .

Optimized geometries of Cu²⁺ active sites adsorption on (a) C₃P, (b) P=O, (c) P(OH)₂, (d) (OH)P=O, respectively. Numbers in black are distances (Å) between atoms.

On this basis, DFT modeling is used to demonstrate the detailed mechanism of acetylene hydrochlorination catalyzed by active copper species on C₃P substrates. Meanwhile, the reaction path in the figure below provides a transition state, which improves the evolution of the reaction and makes the results more convincing.

DFT calculations on the reaction mechanism. (a) Energy profile of acetylene hydrochlorination of HCl and C₂H₂. (b) Optimized geometries of intermediates and transition states. Numbers are distances (Å) between atoms, numbers in red are Hirshfeld charges, and numbers in brown are Mayer bond indices.

Considering the reviewer's suggestion, we rewrite the calculation part. Line 432, the statement of "The adsorption energy of C₂H₂ on five different phosphorus species active sites are -31.77, -42.27, -85.07, -21.26, -16.54 kJ/mol respectively (Fig. S7b-f). It is obvious that P(OH)₂ and (OH)P=O copper active sites show weak interaction with C₂H₂. The P=O active site shows better adsorption to C₂H₂, but O of P=O bond may interact strongly with the Cl of HCl, which may cause complex reaction mechanism. Here we do not discuss the P=O species. The (OH)₂P=O would decompose at our reaction temperature. Then only the C₃P specie was left.

The optimized structure of Cu^{2+} species adsorbed on the C_3P carbon support exist strong interaction between P and Cl, which may be real reason for the activity of our P-doped Cu-based catalyst. After C_2H_2 and HCl adsorbed on the active site, the $\text{C}\equiv\text{C}$ bond length and H-Cl bond length is 1.2254Å and 1.3354Å respectively (Fig. 6a, d), which is 1.2012Å and 1.2895Å in the gas phase in reference. As is shown in the Fig. 6b, c, the molecular orbital of C_2H_2 -complex displays interaction between C_2H_2 and Cu^{2+} active sites, and there is an orbital overlap between the Cu atom d orbital and the acetylene π^* orbital, while the molecular orbital of HCl-complex is different between HCl and active sites. However, the adsorption energy of C_2H_2 and HCl on Cu^{2+} species are calculated at -31.77 and -34.13 kJ/mol. With cooperation of the nearly equal adsorption energy, it renders that the Cu^{2+} species after loaded to the P-doped carbon support can active both C_2H_2 and HCl of the same class. Subsequently, the adsorption of C_2H_2 and HCl at the active site may cause acetylene hydrochlorination reaction following the L-H mechanism shown in Fig. 6e. C_2H_2 and HCl co-adsorbed species (C_2H_2^* and HCl^*) reacted with each other to produce vinyl chloride” was rewritten as “As shown in Fig. S9, the adsorption energies of CuCl_2 with similar configurations on C_3P and $\text{P}=\text{O}$ substrates are -240.5 kJ/mol and -86.38 kJ/mol, respectively. Cl atoms are located above P in C_3P and $\text{P}=\text{O}$, at distances of 2.59 Å and 2.31 Å, respectively. For the $\text{P}(\text{OH})_2$ adsorption geometry, the adsorption energy of CuCl_2 on it is 182.21 kJ/mol, and the Cl ligand in the Cu center seems to interact with the H atom of the hydroxyl group, and the corresponding H-Cl bond length of 2.02 Å. The adsorption energy of CuCl_2 on $(\text{OH})\text{P}=\text{O}$ is -70.89 kJ/mol, and the Cu atom tends

to bond with the O atom in the P=O functional group. The active copper species have the highest adsorption energy on C₃P, indicating a stronger interaction between the Cu center and the substrate. It can be realized that the interaction between the P atom and its nearby Cl atom will affect the potential reaction mechanism of acetylene hydrochlorination. Therefore, the reaction mechanism on the coordination structure formed by CuCl₂ and C₃P is studied.

The mechanism details of acetylene hydrochlorination catalyzed by active copper species on C₃P substrates are presented in Fig. 7 through the DFT modeling. To make the figure clearer, we used simply lines instead of CPK modes to simulate graphene ring by VMD software⁶³. The calculated energy profile is shown in Fig. 7a, and corresponding optimized configurations involved are shown in Fig. 7b. In Fig. 7a, the reaction begins with the coordination of C₂H₂ with Cu, a metal atom of CuCl₂ in active catalyst **a**, forming intermediate **b** with an adsorption energy of -22.32 kJ/mol and a bond length of C≡C of 1.23 Å (the normal bond length of 1.21 Å). Meanwhile, the P-Cl bond length is 2.44 Å, indicating that phosphorus atom in C₃P will have strong electrostatic interaction with Cl atom in Cu-Cl bond, resulting in electron transfer, as evidenced by the Mayer bond index, which will affect the electronic states around Cu atoms and thus affect the adsorption of substrates. The Cu 2p XPS and fitting parameters from the EXAFS spectra of Cu/PC800 all indicate the coordination between Cu and Cl atom, which also verifies the indirect effect of P atoms on Cu. Then formation of C₂H₂ and HCl co-adsorption configuration on catalyst support in **c** shows the H-Cl bond length in HCl is stretched to 1.342 Å slightly longer than the normal bond length of 1.289 Å in free HCl. With

acetylene following adsorbed at the Cl atom of CuCl_2 due to the electrostatic attraction between H atom of C_2H_2 and Cl atom of CuCl_2 to form a weakly less stable intermediate **d**, the distance between the Cl atom of HCl and Cu atom is 3.14 Å and the Mayer bond index is 0.103. Meanwhile, the distance and Mayer bond index of H atom of HCl and the Cl atom of CuCl_2 is 3.14 Å and 0.103. Hirshfeld charges of Cu atom and P atom changes from 0.267 to 0.181 and 0.364 to 0.355 apparently, respectively, which proved our speculate to some extent. Then significantly, H atom of HCl attacks a C atom of acetylene implied with a six-membered ring structure, which consists of HCl, C_2H_2 , and CuCl_2 in **e**. Visibly, H-Cl bond length in HCl tends to be broken (1.64 Å). Meanwhile, the bonding tendency of H atom of HCl and C atom of acetylene is evidenced by the distance and the Mayer bond index, which is 1.30 Å and 0.438, respectively. The Cu center becomes a CuCl_3 coordination structure because of the substitution of Cl in **e**, as evidenced by the change of Hirshfeld charge on Cu center. This step requires an overall activation energy of 59.85 kJ/mol and leads to the product complex **f**. At last, desorption of the chloroethylene molecule from **f** regenerates the catalyst a with weak desorption energy”.

3. In fig 4, it seems that more than one factor shown is positively correlated with catalytic activity, and the authors should specify which is the most vital one.

Response: Thanks for the reviewer’s advice. We are sorry that we may not have made it clear in the manuscript. In Fig. 4, Cu^{2+} and P-C bonds are obviously positively correlated with catalytic activity, and other factors show little change or are negatively correlated with catalytic activity. According to the reviewer’s suggestion,

in line 314 of the manuscript, “ Cu^{2+} species and P-C bond can play a positive role in the hydrochlorination of acetylene, and the coordination structure formed by the interaction between phosphorus species (P-C) and isolated single-atomic Cu^{2+} species is the main active site of the Cu-based catalyst” was added.

4. In addition to XPS, the XANES and soft X-ray absorption spectra should be provided to further illustrate the electronic properties of Cu species.

Response: Thanks to the reviewer for the suggestion, which is very valuable. We have supplemented the X-ray absorption spectra (XAS) of Cu/PC800 to further confirm the oxidation state and precise coordination structure of the Cu element in Cu/PC800. The results show that Cu in Cu/PC800 mainly exists as Cu^{2+} oxidation state and coordinates with Cl atom. Line 396, “X-ray absorption spectroscopy (XAS) further confirms the oxidation state and precise coordination structure of the Cu element in Cu/PC800. The white line height of each sample is shown in the Fig. 6a. The white line intensity values of cationic Cu standard of Cu^{2+} is 1.18, which is close to the measured values in the literature⁶¹. The Cu/PC800 catalyst shows similar spectral features to reference CuCl_2 , and its normalized white line intensity value is 1.17, which is close to the white line intensity of the cationic Cu^{2+} standard sample, indicating that the isolated Cu atoms bear a positive charge of +2. This is consistent with the above XPS results. Although XANES analysis provides relevant information concerning Cu speciation, the complexity of the spectra requires additional extended XAFS (EXAFS) analysis to clarify interpretation. Fig. 6b shows EXAFS Fourier

transforms (FTs) of Cu foil reference, CuCl₂ reference and Cu/PC800. Fourier transformed R-space curves of the Cu K-edge EXAFS spectra suggests that Cu is predominantly coordinated with Cl atom in Cu/PC800 centred at about 2.17 Å, and the average coordination number is 3.8. What is important is that no Cu-Cu characteristic bonds are detected, indicating the atomic distribution of Cu elements in the catalyst, which confirms the atomic dispersion of Cu shown by HAADF-STEM. Fig. 6c indicates that the experimental data of the EXAFS spectrum are well fitted, as shown by the parameter R factor in Table S8” was added.

Table S8 Fitting parameters from the EXAFS spectra of the selected catalysts.[†]

Sample [†]	Scattering Path [†]	CN ^a [†]	R(Å) ^b [†]	$\sigma^2 \times 10^{-3} (\text{Å}^2)$ ^c [†]	R factor (%) [†]
Cu foil [†]	Cu-Cu [†]	12 [†]	2.55±0.01 [†]	8.9±0.5 [†]	0.32 [†]
CuCl ₂ [†]	Cu-Cl [†]	3.7±0.3 [†]	2.16±0.01 [†]	11.0±1.1 [†]	0.82 [†]
Cu/PC800 [†]	Cu-Cl [†]	3.8±0.3 [†]	2.17±0.01 [†]	11.1±1.1 [†]	0.84 [†]

^a Coordination number. ^b Coordination shell distance. ^c Debye-Waller factor.[†]

(a) Cu K-edge-normalized XANES spectra of the sample and reference material. (b) Fourier-transformed magnitude of Cu foil, CuCl₂ and Cu/PC800 (no phase correction). Experimental and fitted EXAFS spectra at the Cu K-edge of the selected catalysts, (c) k-space.

5. According to Cu LMM Auger spectra, Cu⁰ Cu⁺ Cu²⁺ all exists within the catalyst, and this is contradictory to the single atom structure, because Cu⁰ barely exists in single atom catalysts.

Response: Thank you very much for your good comments, which inspired me to think again. The HAADF-STEM images in the manuscript show that Cu active components mostly exist in the form of highly dispersed isolated copper species, we are sorry that the description in the manuscript may be quite absolute, and we have revised it considering the comments of reviewer. Line 187, the statements of “In addition, further analysis of HAADF-STEM image reveals the existence of highly dispersed isolated copper species, and no copper nanoparticles are found. It’s confirmed that the single center copper species supported on carbon support is the active center of acetylene hydrochlorination reaction, indicating that the active component of the catalyst is composed of atom dispersed copper” were corrected as “In addition, further analysis of HAADF-STEM image (Fig. 1d and Fig. S4a, c, e) revealed the presence of predominantly highly dispersed isolated Cu species, and it is almost difficult to detect copper nanoparticles. The single center copper species supported on the carbon support is confirmed to be the active center of acetylene hydrochlorination reaction, indicating that the active component of the catalyst is mostly composed of atomically dispersed copper⁴⁰⁻⁴²”. Cu 2p XPS spectra shows that Cu²⁺ is the main active component. Although Cu⁰ exists, its influence on catalytic performance is relatively small, so it has not been discussed in depth. In addition,

XANES spectra and fitting parameters from the EXAFS spectra of Cu/PC800 shows the presence of Cu-Cl bond, but no Cu-Cu bond is detected, indicating that Cu mainly exists in an isolated form and the concentration of metallic Cu nanoparticles can be ignored, which is consistent with HAADF-STEM results. Therefore, the presence of Cu⁰ in the catalyst may be due to the reduction of copper ions by the AC support (Catalysis Letters (2014) 144:1-8) or by the bombardment of the ion beam during the test (ACS Catalysis (2014) 8: 8493-8505). There are also certain errors in the processing of XPS spectra.

6. The catalysts after reaction should also be characterized to illustrate the stability issue.

Response: The reviewer's suggestion is very meaningful. As the reviewer suggested, we have supplemented the characterization of the reacted catalyst to illustrate its stability, especially for Cu/PC800 with the best catalytic activity. The following is my supplementary content (line 379): The Fig. S8a shows the XRD pattern of the used sample, which is similar to the fresh catalyst. The two main diffraction peaks at approximately 25° and 43° correspond to the (002) and (101) crystal planes of carbon, respectively. Except for the two diffraction peaks, no other characteristic peaks were found. The used Cu/PC800 catalyst with the highest catalytic activity was further characterized. The HAADF-STEM image (Fig. S8b) shows the presence of highly dispersed isolated copper species. The Cu 2p XPS spectrum (Fig. S8c) shows that the peak representing Cu²⁺ at 934.7 eV dominates the spectrum, accounting for 70%. As

mentioned above, according to the nitrogen adsorption and desorption isotherms, the used Cu/PC800 does not have obvious carbon deposition. It can be seen that Cu/PC800 is relatively stable during the reaction, active copper species is not easy to agglomerate and not easy to reduce, and the catalyst also has a certain ability to resist carbon deposition.

(a) XRD pattern of used P-doped Cu-based catalysts, (b) Representative HAADF-STEM image and (c) Cu 2p XPS spectra of the used Cu/PC800.

7. Following references regarding atomically dispersed catalysts should be cited.
Journal of the American Chemical Society, 2021, 143, 1, 309–317; *ACS Catalysis*, 2020, 10, 907-913; *Nature Nanotechnology*, 2019, 14, 354–361.; *Journal of the American Chemical Society*, 2019, 141, 18921-18925.

Response: Special thanks to you for your good comments. The following literatures regarding atomically dispersed catalyst has certain reference value for this manuscript. We have read these literatures and cited them in the manuscript. As follows:

65. Lin, L. et al. Atomically dispersed Ni/ α -MoC catalyst for hydrogen production from methanol/water. *J. Am. Chem. Soc.* **143**, 309-317 (2021).

66. Li, S. et al. Impact of the coordination environment on atomically dispersed Pt catalysts for oxygen reduction reaction. *ACS Catal.* **10**, 907-913 (2020).

67. Lin, L. et al. A highly CO-tolerant atomically dispersed Pt catalyst for chemoselective hydrogenation. *Nat. Nanotechnol.* **14**, 354-361 (2019).

68. Jin, R. et al. Low temperature oxidation of ethane to oxygenates by oxygen over iridium-cluster catalysts. *J. Am. Chem. Soc.* **141**, 18921-18925 (2019).

Thank you again for your suggestions. All your suggestions are very important, and they have important guiding significance for my future scientific research work.

Reviewer #3:

In the process of carrier doping with phosphorus, the author adjusts the types of phosphorus species and their proportions on the surface of the carrier through different calcination temperatures. Combined with the activity, it is determined that the phosphorus species (P-C) plays an important role in the hydrochlorination of acetylene. The interaction between this phosphorus species and atomically dispersed Cu species is beneficial to catalytic activity. Further calculations of density functional theory (DFT) show how the reactants are activated to play a catalytic role. This is a carefully done study and the findings are innovative and insightful. This is fine, because it is already known that atomic dispersed metal sites coordinated on P-doped carbon displayed outstanding catalytic performance and reaction generality in the heterogeneous catalytic reactions (Nature Communications (2020) 11:4074), but little literature has been reported on

achieving precise and atomic control to prepare Cu single-atom catalysts for hydrochlorination of acetylene, with excellent catalytic performance as well. In this work, the experimental preparation strategy is ingenious and the mechanism is enlightening, which can certainly contribute to the knowledge of single-atom catalysts for acetylene hydrogenation. This manuscript is recommended for publication after minor revision.

Response: The paper mentioned by the reviewer (Nature Communications (2020) 11: 4074) illustrates the excellent catalytic performance of atomic dispersed metal sites coordinated on P-doped carbon in heterogeneous catalytic reactions, which is of reference significance to this paper. We have read and cited this literature. As follows:

64. Long, X. et al. Graphitic phosphorus coordinated single Fe atoms for hydrogenative transformations. *Nat. Commun.* **11**, 4074 (2020).

1. How the coordination structure formed by the P-C bond formed after the phosphorus element is incorporated into the carbon lattice and the copper species activates the substrates? The description of this part of the article is too simple, you can describe it in detail.

Response: Thanks for the suggestion of reviewer. I'm really sorry that we didn't write in detail in this section. We have made modifications according to the comment of the reviewer. Our original substrate C₁₃H₉ refers to the literature (RSC Advances (2015) 5: 104071-104078), but the original substrate may be too simple, so we refer to the updated literature again (Inorganic Chemistry Frontiers (2019) 6: 2944-2952), a larger and more realistic substrate is selected for re-calculation during modeling, as shown

in the figure below. By DFT calculation, when the phosphorus atom replaces a carbon atom in the carbon skeleton and presents a C₃P configuration, the energy released when it interacts with Cu²⁺ species to form coordination structure is the largest, indicating that the interaction between active copper species and phosphorus-doped carbon carrier is the strongest. Therefore, we have obtained that phosphorus and carbon are coordinated by C₃P.

Optimized geometries of Cu²⁺ active sites adsorption on (a) C₃P, (b) P=O, (c) P(OH)₂, (d) (OH)P=O, respectively. Numbers in black are distances (Å) between atoms.

On this basis, DFT modeling is used to demonstrate the detailed mechanism of acetylene hydrochlorination catalyzed by active copper species on C₃P substrates. Meanwhile, the reaction path in the figure below provides a transition state, which improves the evolution of the reaction and makes the results more convincing.

DFT calculations on the reaction mechanism. (a) Energy profile of acetylene hydrochlorination of HCl and C_2H_2 . (b) Optimized geometries of intermediates and transition states. Numbers are distances (Å) between atoms, numbers in red are Hirshfeld charges, and numbers in brown are Mayer bond indices.

Considering the reviewer's suggestion, we rewrite the calculation part. Line 432, the statement of "The adsorption energy of C_2H_2 on five different phosphorus species active sites are -31.77, -42.27, -85.07, -21.26, -16.54 kJ/mol respectively (Fig. S7b-f). It is obvious that $P(OH)_2$ and $(OH)P=O$ copper active sites show weak interaction with C_2H_2 . The $P=O$ active site shows better adsorption to C_2H_2 , but O of $P=O$ bond may interact strongly with the Cl of HCl, which may cause complex reaction mechanism. Here we do not discuss the $P=O$ species. The $(OH)_2P=O$ would decompose at our reaction temperature. Then only the C_3P specie was left.

The optimized structure of Cu^{2+} species adsorbed on the C_3P carbon support exist strong interaction between P and Cl, which may be real reason for the activity of our P-doped Cu-based catalyst. After C_2H_2 and HCl adsorbed on the active site, the $\text{C}\equiv\text{C}$ bond length and H-Cl bond length is 1.2254Å and 1.3354Å respectively (Fig. 6a, d), which is 1.2012Å and 1.2895Å in the gas phase in reference. As is shown in the Fig. 6b, c, the molecular orbital of C_2H_2 -complex displays interaction between C_2H_2 and Cu^{2+} active sites, and there is an orbital overlap between the Cu atom d orbital and the acetylene π^* orbital, while the molecular orbital of HCl-complex is different between HCl and active sites. However, the adsorption energy of C_2H_2 and HCl on Cu^{2+} species are calculated at -31.77 and -34.13 kJ/mol. With cooperation of the nearly equal adsorption energy, it renders that the Cu^{2+} species after loaded to the P-doped carbon support can active both C_2H_2 and HCl of the same class. Subsequently, the adsorption of C_2H_2 and HCl at the active site may cause acetylene hydrochlorination reaction following the L-H mechanism shown in Fig. 6e. C_2H_2 and HCl co-adsorbed species (C_2H_2^* and HCl^*) reacted with each other to produce vinyl chloride” was rewritten as “As shown in Fig. S9, the adsorption energies of CuCl_2 with similar configurations on C_3P and $\text{P}=\text{O}$ substrates are -240.5 kJ/mol and -86.38 kJ/mol, respectively. Cl atoms are located above P in C_3P and $\text{P}=\text{O}$, at distances of 2.59 Å and 2.31 Å, respectively. For the $\text{P}(\text{OH})_2$ adsorption geometry, the adsorption energy of CuCl_2 on it is 182.21 kJ/mol, and the Cl ligand in the Cu center seems to interact with the H atom of the hydroxyl group, and the corresponding H-Cl bond length of 2.02 Å. The adsorption energy of CuCl_2 on $(\text{OH})\text{P}=\text{O}$ is -70.89 kJ/mol, and the Cu atom tends

to bond with the O atom in the P=O functional group. The active copper species have the highest adsorption energy on C₃P, indicating a stronger interaction between the Cu center and the substrate. It can be realized that the interaction between the P atom and its nearby Cl atom will affect the potential reaction mechanism of acetylene hydrochlorination. Therefore, the reaction mechanism on the coordination structure formed by CuCl₂ and C₃P is studied.

The mechanism details of acetylene hydrochlorination catalyzed by active copper species on C₃P substrates are presented in Fig. 7 through the DFT modeling. To make the figure clearer, we used simply lines instead of CPK modes to simulate graphene ring by VMD software⁶³. The calculated energy profile is shown in Fig. 7a, and corresponding optimized configurations involved are shown in Fig. 7b. In Fig. 7a, the reaction begins with the coordination of C₂H₂ with Cu, a metal atom of CuCl₂ in active catalyst **a**, forming intermediate **b** with an adsorption energy of -22.32 kJ/mol and a bond length of C≡C of 1.23 Å (the normal bond length of 1.21 Å). Meanwhile, the P-Cl bond length is 2.44 Å, indicating that phosphorus atom in C₃P will have strong electrostatic interaction with Cl atom in Cu-Cl bond, resulting in electron transfer, as evidenced by the Mayer bond index, which will affect the electronic states around Cu atoms and thus affect the adsorption of substrates. The Cu 2p XPS and fitting parameters from the EXAFS spectra of Cu/PC800 all indicate the coordination between Cu and Cl atom, which also verifies the indirect effect of P atoms on Cu. Then formation of C₂H₂ and HCl co-adsorption configuration on catalyst support in **c** shows the H-Cl bond length in HCl is stretched to 1.342 Å slightly longer than the normal bond length of 1.289 Å in free HCl. With

acetylene following adsorbed at the Cl atom of CuCl_2 due to the electrostatic attraction between H atom of C_2H_2 and Cl atom of CuCl_2 to form a weakly less stable intermediate **d**, the distance between the Cl atom of HCl and Cu atom is 3.14 Å and the Mayer bond index is 0.103. Meanwhile, the distance and Mayer bond index of H atom of HCl and the Cl atom of CuCl_2 is 3.14 Å and 0.103. Hirshfeld charges of Cu atom and P atom changes from 0.267 to 0.181 and 0.364 to 0.355 apparently, respectively, which proved our speculate to some extent. Then significantly, H atom of HCl attacks a C atom of acetylene implied with a six-membered ring structure, which consists of HCl, C_2H_2 , and CuCl_2 in **e**. Visibly, H-Cl bond length in HCl tends to be broken (1.64 Å). Meanwhile, the bonding tendency of H atom of HCl and C atom of acetylene is evidenced by the distance and the Mayer bond index, which is 1.30 Å and 0.438, respectively. The Cu center becomes a CuCl_3 coordination structure because of the substitution of Cl in **e**, as evidenced by the change of Hirshfeld charge on Cu center. This step requires an overall activation energy of 59.85 kJ/mol and leads to the product complex **f**. At last, desorption of the chloroethylene molecule from **f** regenerates the catalyst a with weak desorption energy”.

2. The description of FT-IR in Fig. 3c is not clear enough. It is better to clearly indicate the wavelength position of each functional group and the corresponding literature source.

Response: Thanks to the reviewer for this careful suggestion. We have marked the wavelength positions of each functional group and the corresponding literature sources in the description of Fig. 3c of the manuscript. Line 276, “In the Fig. 3b, the

strong broadband of the four samples in the range of 3500-3200 cm^{-1} corresponds to the -OH stretching vibration, and the broadband around 1300 cm^{-1} can be attributed to the C-O stretching vibration³⁰. The spectrum clearly shows that the peaks at about 1100 cm^{-1} and about 1000 cm^{-1} are attributed to the stretching vibration of P=O and P-O, respectively, and some weaker peaks appearing at 750-660 cm^{-1} are attributed to P-C^{38,50}” was added.

3. The author can put the structure of HEDP in the supporting information so that the reader can judge the possible structure of the phosphorus species in the carbon carrier.

Response: Thanks to the reviewer for the suggestion. Considering the reviewer’s suggestion, we added the structure of HEDP to the supporting information so that readers can judge the possible structure of the phosphorus species in the carbon support.

The structure of HEDP.

Thank you again for your suggestions. All your suggestions are very important, and they have important guiding significance for my future scientific research work.

We hope the Reviewers will be satisfied with the revisions for the original manuscript.

Thanks and Best regards!

Yours Sincerely,

Jia Zhao

jjazhao@zjut.edu.cn (J. Zhao)

REVIEWERS' COMMENTS:

Reviewer #1 (Remarks to the Author):

[Editorial note: The reviewer has provided their comments in the attached file.]

Reviewer #3 (Remarks to the Author):

I reviewed the response letter and revised manuscript, and found all my concerns and the questions from reviewer#2 had been properly addressed. Therefore, I recommend to accept this work without further revision.

Comments

In the manuscript, the authors studied the P doped carbon load single atom copper catalyst in acetylene hydrochlorination reaction, the application of inspected the catalytic performance of the catalyst in the reaction, and the role of the catalysts has carried on the detailed research, not only through the characterization of explain the structure-activity relationship of catalyst in the reaction, and through the simulation study of its mechanism of action, It provides a new reference for the application of Cu-based catalyst in acetylene hydrochlorination. However, there are still some problems that need further modification and improvement.

- 1、 Is the analysis of nitrogen adsorption/desorption isothermal data tested using microporous analysis methods?
- 2、 The EDS picture is not clear enough. If there is a clearer picture, please replace the unclear one in the article.
- 3、 The content of elements in XPS and EDS is not accurate, so it is suggested to give the content through ICP.
- 4、 Please confirm that the serial number of the figure in the supporting literature is consistent with that in the manuscript.
- 5、 There is a problem with the statement in line 193 on page 10. How can XPS results reflect element dispersion?
- 6、 In Figure 2 on page 11, (a) and (b) can be combined into one picture. Please distinguish between the two samples in (d) with the same name "Cu/SAC".
- 7、 The existence of P-O-Cu was mentioned in XPS oxygen element analysis, while the coordination atom of Cu in the simulation was chlorine atom. Please give relevant explanations.
- 8、 On page 24, line 462, check that the values described are consistent with the simulation results.

Dear Reviewers,

On behalf of my co-authors, we thank you very much for giving us an opportunity to revise our manuscript, we appreciate you very much for your positive and constructive comments and suggestions on our manuscript entitled "Isolated Single-Atomic Cu Catalyst Supported on P-doped Carbon for Hydrochlorination of Acetylene". (Manuscript ID: COMMSCHEM-21-0186). We have studied the valuable comments from reviewers carefully, and tried our best to revise the manuscript. The point-to-point response to your comments are listed as following:

Comments in **bold** - Response in black - Actions in yellow

Indicated figures and page numbers refer to the revised manuscript with changes highlighted

Reviewers' comments:

Reviewer #1:

In the manuscript, the authors studied the P doped carbon load single atom copper catalyst in acetylene hydrochlorination reaction, the application of inspected the catalytic performance of the catalyst in the reaction, and the role of the catalysts has carried on the detailed research, not only through the characterization of explain the structure-activity relationship of catalyst in the reaction, and through the simulation study of its mechanism of action, It provides a new reference for the application of Cu-based catalyst in acetylene hydrochlorination. However, there are still some problems that need further modification and improvement.

1. Is the analysis of nitrogen adsorption/desorption isothermal data tested using microporous analysis methods?

Response: Our data was analyzed using microporous analysis. First, in order to eliminate the interference factors generated in the measurement process, the sample needs to be degassed to remove the water in the channel and the impurities on the sample surface. 0.15 g samples were weighed and degassed for 10 h under the vacuum condition of $T = 150\text{ }^{\circ}\text{C}$. After degassing, the sample was cooled to 77 K with liquid nitrogen for physical adsorption. By N_2 adsorption/desorption experiment, we can get the difference of specific surface area and pore structure of catalysts calcination at different temperatures. What we found was that compared with the specific surface area of activated carbon, both the specific surface area and pore volume of activated carbon after phosphorus doping treatment are smaller. The addition of phosphorus element may fill and block part of the pore of the carrier and occupy some available space. The larger specific surface area and pore volume of the catalysts calcined at $600\text{ }^{\circ}\text{C}$ and $800\text{ }^{\circ}\text{C}$ may be caused by the thermal decomposition of phosphorus ligand at high temperature and the reduction of blocked pores. The pore size of phosphorus-doped activated carbon is similar, and it is also relatively close to activated carbon. Previous reports generally agree that a higher specific surface area can expose more active sites to promote the transfer of the substrate, which is conducive to improving the activity. The results of this study are also the same. The carbon carrier calcined at 800°C has the largest specific surface area, and the corresponding acetylene conversion rate is also the highest among several catalysts.

Supplementary Table 2. Textural properties of the phosphorus-doped carbon materials.

Sample	S_{BET} ($\text{m}^2 \text{g}^{-1}$) ^a	Volume ($\text{cm}^3 \text{g}^{-1}$) ^b	Diameter (nm) ^c
AC	1204	0.61	2.03
PC200	186	0.12	2.40
PC400	203	0.13	2.36
PC600	910	0.47	2.04
PC800	1005	0.58	2.19

^a measured using the Brunauer-Emmett-Teller (BET) method; ^b calculated dependent on the adsorbed N_2 volume; ^c determined by Barrett-Joyner-Halenda (BJH) method.

2. The EDS picture is not clear enough. If there is a clearer picture, please replace the unclear one in the article.

Response: Thanks for the reviewer's suggestion. We have replaced the original unclear pictures with new clear one according to the reviewer's suggestion.

EDS elemental mapping of fresh Cu/PC200.

EDS elemental mapping of fresh Cu/PC400.

EDS elemental mapping of fresh Cu/PC600.

EDS elemental mapping of fresh Cu/PC800.

3. *The content of elements in XPS and EDS is not accurate, so it is suggested to give the content through ICP.*

Response: Thank you very much for the suggestion made by this reviewer. As the reviewer said, the content of elements in XPS and EDS is a semi-quantitative result, not very accurate, so we added the ICP-AES data under the reviewer's suggestion. Because ICP-AES can only measure metal elements, other elements cannot be measured or the measurement results are inaccurate, so we only supplemented the data of copper element. The copper content in ICP-AES data is similar to XPS and EDS data, confirming the reliability of the data, indicating that the copper element in the catalyst is well loaded on the support. Line 90, the statement of “The XPS and EDS results listed in the Table S1 and Fig. 1a show that there is a certain amount of phosphorus in this batch of catalysts, indicating that phosphorus is doped in the carbon framework” were corrected as “The XPS and EDS results listed in the Supplementary Table 1, Fig. 1a show that there are a certain amount of copper and phosphorus in this batch of catalysts, indicating that copper is well loaded on the support, and phosphorus is successfully doped in the carbon framework. The copper content in ICP-AES data is similar to XPS and EDS data, which confirms the reliability of the data (Supplementary Table 2)”.

Supplementary Table 1. C, Cu, P and O contents determined by XPS and EDS analysis over different samples.

Sample	XPS				EDS			
	(wt.%)				(wt.%)			
	C	P	Cu	O	C	P	Cu	O
Cu/PC200	51.00	13.20	8.17	27.63	55.51	12.06	10.41	22.02
Cu/PC400	46.00	13.81	9.17	31.02	54.04	12.38	8.14	25.44
Cu/PC600	56.72	11.58	8.39	23.32	57.06	11.81	9.88	21.25
Cu/PC800	76.51	5.25	9.33	8.91	74.95	6.32	10.37	8.36

Supplementary Table 2. ICP-AES analysis of copper in the catalysts and their activities in acetylene hydrochlorination.

Element	Cu/PC200	Cu/PC400	Cu/PC600	Cu/PC800
Cu	8.04%	8.87%	9.31%	8.73%

4. Please confirm that the serial number of the figure in the supporting literature is consistent with that in the manuscript.

Response: Thank you very much for your suggestions. We have rechecked to ensure that the serial number of the figure in the supporting literature is consistent with that in the manuscript.

5. There is a problem with the statement in line 193 on page 10. How can XPS results reflect element dispersion?

Response: The reviewer is very careful, and the proposed suggestion is very reasonable.

I am very sorry that we have not expressed clearly. The original meaning is that the element mapping shows uniformly distributed Cu and P elements, which, together with the result of full XPS spectra in Fig. 1a, indicates the successful doping of phosphorus elements. To avoid ambiguity, line 191, the statements of “The element mapping of the catalyst Cu/PC800 (Fig. 1e) reveals that C, P and Cu elements are uniformly distributed on the surface of the catalyst, which is also consistent with the previous XPS results, verifying the successful doping of phosphorus in the carbon support” were corrected as “The element mapping of the catalyst Cu/PC800 (Fig. 1e) reveals that C, P and Cu elements are uniformly distributed on the surface of the catalyst, verifying the successful doping of phosphorus in the carbon support”.

6. In Figure 2 on page 11, (a) and (b) can be combined into one picture. Please distinguish between the two samples in (d) with the same name "Cu/SAC".

Response: Thank you very much for your suggestions. We have made modifications according to the suggestions of reviewer. (a) and (b) in Figure 2 have been combined into one picture, and two samples with the same name "Cu/SAC" in (d) have been distinguished.

Fig. 2. (a) The conversion of acetylene over P-doped Cu-based catalysts. Reaction conditions: temperature = 150 °C, GHSV(C₂H₂) = 90 h⁻¹, V(HCl)/V(C₂H₂) = 1.2/1; Comparison of acetylene conversions for Cu/PC200, Cu/PC400, Cu/PC600 and Cu/PC800 catalysts and their respective treated carbons, (b) Kinetic studies of Cu/AC and Cu/PC800 catalyst: apparent activation energy, kJ mol⁻¹, (c) GHSV plotted against the Productivity for some copper-based catalysts reported in literature and Cu/PC800 catalyst with better catalytic performance in this article.

7. The existence of P-O-Cu was mentioned in XPS oxygen element analysis, while the coordination atom of Cu in the simulation was chlorine atom. Please give relevant explanations.

Response: The reviewer noted this problem with great care. Thank the reviewer for pointing out this problem, we think it is very reasonable. XPS data analysis may be subjective to some extent. By referring to the literature, as shown in the figure below, the peak around 532.5eV may be P-O-Cu (Applied Catalysis A: General (2020) 591: 117408) or C-O-P (ACS Nano (2016) 10: 2305-2315).

O 1s XPS spectra.

O 1s XPS spectra.

We are very sorry that we did not notice the problem raised by the reviewer at the beginning. The peak around 532.5eV was defined as P-O-C subjectively and was not combined with the subsequent results of XAS. Thanks for the suggestion of the reviewer, which made us reconsider. Considering that the coordination atom of Cu in XANES spectra is Cl atom, it would be more suitable for the peak about 532.5eV to be attributed to C-O-P. We have modified it in the relevant figure (Fig. 3f and Fig. S7) and article.

Fig. 3f O 1s XPS spectra of fresh P-doped Cu-based catalysts.

Fig. S7 Correlation diagram of (a-b) oxygen species content and productivity.

Line 296, the statements of “It further indicates that P-O-Cu bond is included in the oxygen related phosphorus-containing functional group represented by P-O bond, and the coordination structure of Cu and O in P-O bond is relatively stable” were corrected as “It further indicates that P-O-C bond is included in the oxygen related phosphorus-containing functional group represented by P-O bond”. The corresponding “P-O-Cu” in lines 294-306 has been modified to “P-O-C”.

8. On page 24, line 462, check that the values described are consistent with the simulation results.

Response: Many thanks to the reviewer for finding this problem carefully. We have re-compared the figure and the text and made modifications. Line 460, the statements of “Meanwhile, the distance and Mayer bond index of H atom of HCl and the Cl atom of CuCl₂ is 3.14 Å and 0.103” were corrected as “Meanwhile, the distance and Mayer bond index of Cl atom of HCl and the Cu atom of CuCl₂ is 3.14 Å and 0.103”.